# LIPG signaling promotes tumor initiation and metastasis of human basal-like triple-negative breast cancer

Pang-Kuo Lo[1], Yuan Yao[1], Ji Shin Lee[2], Yongshu Zhang[1], Weiliang Huang[3], Maureen A Kane[3], Qun Zhou[1]*

[1]Department of Biochemistry and Molecular Biology, Greenebaum Cancer Center, University of Maryland School of Medicine, Baltimore, United States; [2]Department of Pathology, Chonnam National University Medical School, Gwangju, Korea; [3]Department of Pharmaceutical Sciences, University of Maryland School of Pharmacy, Baltimore, United States

**Abstract** Current understanding of aggressive human basal-like triple-negative breast cancer (TNBC) remains incomplete. In this study, we show endothelial lipase (LIPG) is aberrantly overexpressed in basal-like TNBCs. We demonstrate that LIPG is required for *in vivo* tumorigenicity and metastasis of TNBC cells. LIPG possesses a lipase-dependent function that supports cancer cell proliferation and a lipase-independent function that promotes invasiveness, stemness and basal/epithelial-mesenchymal transition features of TNBC. Mechanistically, LIPG executes its oncogenic function through its involvement in interferon-related DTX3L-ISG15 signaling, which regulates protein function and stability by ISGylation. We show that DTX3L, an E3-ubiquitin ligase, is required for maintaining LIPG protein levels in TNBC cells by inhibiting proteasome-mediated LIPG degradation. Inactivation of LIPG impairs DTX3L-ISG15 signaling, indicating the existence of DTX3L-LIPG-ISG15 signaling. We further reveal LIPG-ISG15 signaling is lipase-independent. We demonstrate that DTX3L-LIPG-ISG15 signaling is essential for malignancies of TNBC cells. Targeting this pathway provides a novel strategy for basal-like TNBC therapy.
DOI: https://doi.org/10.7554/eLife.31334.001

*For correspondence:
qzhou@som.umaryland.edu

Competing interests: The authors declare that no competing interests exist.

## Introduction

Breast cancer is a heterogeneous disease and has been classified into several molecular subtypes, including basal-like, HER2-positive, and luminal breast cancers (*Perou et al., 2000*; *Sørlie et al., 2001*). The majority of diagnosed basal-like breast cancer is triple-negative breast cancer (TNBC), which is characterized by its lack of estrogen (ER), progesterone (PR), and HER2 receptors, and contributes to the lack of targeted therapies for basal-like breast cancer (*Rakha et al., 2009*). Basal-like TNBC is highly aggressive, with poor clinical outcomes due to its high tumor grade, increased rate of proliferation and metastasis, and frequent recurrence (*Sørlie et al., 2001*; *Sorlie et al., 2003*; *Badve et al., 2011*). Basal-like TNBC is highly associated with the epithelial-mesenchymal transition (EMT) (*Lehmann et al., 2011*). Moreover, basal-like TNBCs are enriched for poorly differentiated cancer stem cells (CSCs) with the CD24-CD44+ phenotype (*Honeth et al., 2008*). These EMT and CSC features contribute to aggressiveness, metastasis and chemoresistance of basal-like TNBC. Nevertheless, what pathological factors confer these features on basal-like TNBC remain largely unknown.

Cancer cells are known to manifest metabolic dysregulation (e.g. the Warburg Effect) (*Koppenol et al., 2011*). In addition to deregulation of carbohydrate metabolism, dysregulation of lipid uptake, synthesis and metabolism has been found in various types of cancer and some of these

alterations have been shown to be crucial for tumorigenesis (*Santos and Schulze, 2012*; *Baumann et al., 2013*; *Beloribi-Djefaflia et al., 2016*; *Slebe et al., 2016*). Lipoprotein lipases belong to the triglyceride (TG) lipase gene family and play a crucial role in the metabolism and cellular uptake of plasma lipoproteins (*McCoy et al., 2002*). Lipoprotein lipases enzymatically digest lipoproteins to liberate their lipid contents (*McCoy et al., 2002*). The released lipid precursors can be taken up by the cell for incorporation into lipid synthesis or lipid metabolism to generate energy (*Slebe et al., 2016*). In addition, lipoprotein lipases promote the binding and cellular uptake of lipoproteins in an enzyme-independent manner (*Strauss et al., 2002*). Aberrant lipoprotein lipase expression has been reported to contribute to lipid dysregulation in several types of cancer and promote tumorigenesis by supplying cancer cells with lipid precursors, a high-energy fuel for cancer cell proliferation (*Nielsen et al., 2010*; *Slebe et al., 2016*; *Cadenas et al., 2012*). However, whether and how lipoprotein lipase dysregulation impacts on other malignant features (e.g. invasiveness, stemness, EMT) remain uncharacterized.

The interferon-stimulated gene 15 (ISG15) encodes an interferon (IFN)-inducible, ubiquitin-like protein. In cells, ISG15 is conjugated to an array of proteins (a process called ISGylation) functionally implicated in various cellular processes, encompassing IFN-induced immune responses and the modulation of cellular protein turnover (e.g. Ubiquitin/26S Proteasome Pathway) (*Desai et al., 2006*; *Hermann and Bogunovic, 2017*). Growing studies show that ISG15 expression and protein ISGylation are dysregulated in numerous types of cancer and play a crucial role in tumorigenesis (*Sgorbissa and Brancolini, 2012*). For example, aberrant overexpression of ISG15 has been shown to promote breast cancer cell invasion and tumorigenicity by disrupting cytoskeletal architecture, promoting cellular motility and enhancing the oncogenic function of Ki-Ras (*Hadjivasiliou, 2012*; *Burks et al., 2014*). Elevated ISG15 expression has been reported in mouse embryonic fibroblasts (MEFs) with co-inactivation of both tumor suppressor genes p53 and ARF and is required for tumorigenic phenotypes exhibited in transformed MEFs (*Forys et al., 2014*). Given that TNBCs frequently displayed defects in both p53 and ARF and over half of TNBCs overexpressed ISG15 (*Forys et al., 2014*), these findings suggest the potential role of ISG15 in TNBC development and progression.

Deltex (DTX)−3-like E3 ubiquitin ligase (DTX3L/BBAP) gains attention due to its roles in immune responses, DNA damage signaling and tumorigenesis. IFN-β and IFN-γ have been shown to induce DTX3L expression, suggesting its physiological involvement in IFN inflammatory signaling (*Juszczynski et al., 2006*; *Zhang et al., 2015b*). Moreover, DTX3L regulates DNA damage repair by promoting the recruitment of BRCA1 and 53BP1 to DNA damage sites (*Yan et al., 2009*; *Yan et al., 2013*). DTX3L has been recently linked to cancer due to its role in promoting metastasis of melanoma via activation of the FAK/PI3K/AKT signaling pathway (*Thang et al., 2015*). DTX3L is reported to positively regulate the expression of IFN-stimulated genes (ISGs), including ISG15, through monoubiquitination of a subset of histones (*Zhang et al., 2015b*). Given the increasing importance of both DTX3L and ISG15 in tumorigenic regulation, the DTX3L-ISG15 regulatory axis may play a pivotal role in cancer development and progression, and deserves further study in TNBC.

In this study, we have found that compared to other breast cancer subtypes, basal-like TNBCs preferentially overexpress endothelial lipase (LIPG), a member of the TG lipase gene family. Our studies further show that LIPG overexpression promotes the development and metastasis of basal-like TNBC through its functional involvement in the oncogenic DTX3L-ISG15 signaling pathway. Moreover, our findings show that LIPG possesses both lipase-dependent and lipase-independent functions to promote the different aspects of basal-like TNBC malignancy. Our findings provide novel insights into the oncogenic role of LIPG in breast cancer and its functional link to IFN-related signaling.

## Results

### LIPG is preferentially overexpressed in basal-like TNBCs

To explore the roles of lipoprotein lipases in breast cancer, we performed *in silico* gene expression analysis of lipoprotein lipases, including lipoprotein lipase (*LPL*), hepatic lipase C (*LIPC*) and endothelial lipase G (*LIPG*), in breast cancer using the Oncomine cancer gene expression database (https://www.oncomine.org) (*Rhodes et al., 2007*). We found that both LPL and LIPC were underexpressed in invasive ductal carcinomas (IDCs) and were not differentially expressed among distinct molecular

breast cancer subtypes (*Figure 1—figure supplement 1* and *Figure 1—figure supplement 2*). In contrast, LIPG expression was significantly elevated in TNBCs when compared to normal breast, HER2-positive and ER-positive luminal IDCs (*Figure 1A*) according to *in silico* analysis of The Cancer Genome Atlas (TCGA) dataset. Consistently, analysis of the Gluck dataset (*Glück et al., 2012*) showed that overall LIPG was expressed at a higher level in basal-like breast cancers (BLBC) than in luminal-A/B breast cancers (*Figure 1B*). These *in silico* analyses suggest a potential role of LIPG in basal-like TNBC.

To validate these *in silico* analyses, we performed qRT-PCR analysis of LIPG mRNA expression in normal breast tissues (n = 6), TNBCs, and luminal breast cancers (LuBCs) positive for ER and PR markers (n = 8 for each subtype). Consistently, the data showed that LIPG mRNA expression was significantly higher in TNBCs compared to normal breast tissues and LuBCs (*Figure 1C*). To further confirm the qRT-PCR data, LIPG protein expression was determined by immunohistochemistry (IHC) in the same cohort of breast tumor tissue samples. To expand the breast cancer tissue cohorts of LuBC and TNBC in IHC studies, we also performed IHC analysis of tissue microarrays encompassing 100 breast cancer cases. A total of 66 breast cancer cases on tissue microarrays were successfully stained by the LIPG antibody (*Figure 1—figure supplement 3*), including 47 LuBC and 10 TNBC cases. By combining the microarray tissue cohort with the aforementioned tissue cohort, IHC staining results of 55 LuBC and 18 TNBC cases were subjected to H-score analysis (*Figure 1D*). In line with LIPG mRNA expression data, LIPG protein was detected at a higher level in TNBCs than in LuBCs according to H-score analysis of IHC-stained samples (*Figure 1D* and *Figure 1—figure supplement 4*).

To examine LIPG expression in breast cancer cell lines, we performed *in silico* analysis of LIPG mRNA expression in the Hoeflich dataset (*Hoeflich et al., 2009*) retrieved from the Oncomine database, including microarray expression datasets of 48 different breast cancer cell lines [basal-like (n = 20), Her2-amplified (n = 15) and luminal (n = 13)]. Consistent with results from breast cancer tissues, LIPG was expressed at a higher level in basal-like cell lines when compared to Her2-amplified and luminal cell lines (*Figure 1E*). To confirm the *in silico* analysis result, we performed western blot analysis of LIPG protein expression on four basal-like TNBC (MCF10DCIS (*Miller et al., 2000*; *Hu et al., 2008*), MDA-MB-231, MDA-MB-468 and Hs578T) and two LuBC (T47D and MCF7) cell lines. We included a LIPG-overexpressing MCF7 cell line (MCF7-LIPG) as a positive control in the experiment. LIPG protein has been detected at 68 kDa and 40 kDa in cell lysates, which represent glycosylated full-length LIPG and cleaved LIPG protein, respectively (*Edmondson et al., 2009*). As shown in *Figure 1F*, high 68 kDa LIPG expression was detected in TNBC, but not in LuBC cell lines, and high 40 kDa LIPG expression was only detected in Hs578T cells. These data indicate that glycosylated, full-length LIPG protein is the predominant form expressed in TNBC cell lines.

## LIPG overexpression confers increased basal/EMT, migratory and cancer stem cell features on luminal cancer cells

To reveal the functional role of LIPG in breast cancer, we created the LIPG-overexpressing MCF7 cell line (MCF7-LIPG) whose parental cells lack LIPG expression (*Figure 1F*). To assess the effect of LIPG overexpression on cell growth, we compared the growth rates of parental MCF7 and MCF7-LIPG with or without LIPG knockdown by siRNA. Western blot analysis was performed to confirm LIPG overexpression in MCF7-LIPG cells and the efficiency of LIPG siRNA in knockdown of both 68 kDa and 40 kDa LIPG proteins in MCF7-LIPG cells (*Figure 2—figure supplement 1*). As expected, LIPG knockdown had no impact on the growth of parental MCF7 cells (MCF7-CTRL) with LIPG deficiency (*Figure 2A*). However, although MCF7 cell growth was not significantly affected by LIPG overexpression during the 3-day growth period, LIPG knockdown dramatically inhibited the growth of MCF7-LIPG cells (*Figure 2A*). These results suggest that the growth of MCF7 cells becomes reliant on LIPG after overexpression of this lipase gene.

To examine whether LIPG regulates cell motility and stemness, we performed migration and stem-cell sphere assays on parental MCF7-CTRL and MCF7-LIPG with or without LIPG knockdown. The data showed that LIPG overexpression promoted migration and primary as well as secondary CSC sphere formation of MCF7 cells, whereas these enhanced aggressive phenotypes reverted to their parental phenotypes after LIPG knockdown (*Figure 2B and C*). We also analyzed the CD24/CD44 profiles of these four different cell samples. The percentage of CD24-/CD44+ cells representing the CSC-enriched population doubled in MCF7-LIPG cells compared to parental control cells, and LIPG knockdown reverted the CSC population back to the baseline value (*Figure 2D*). To

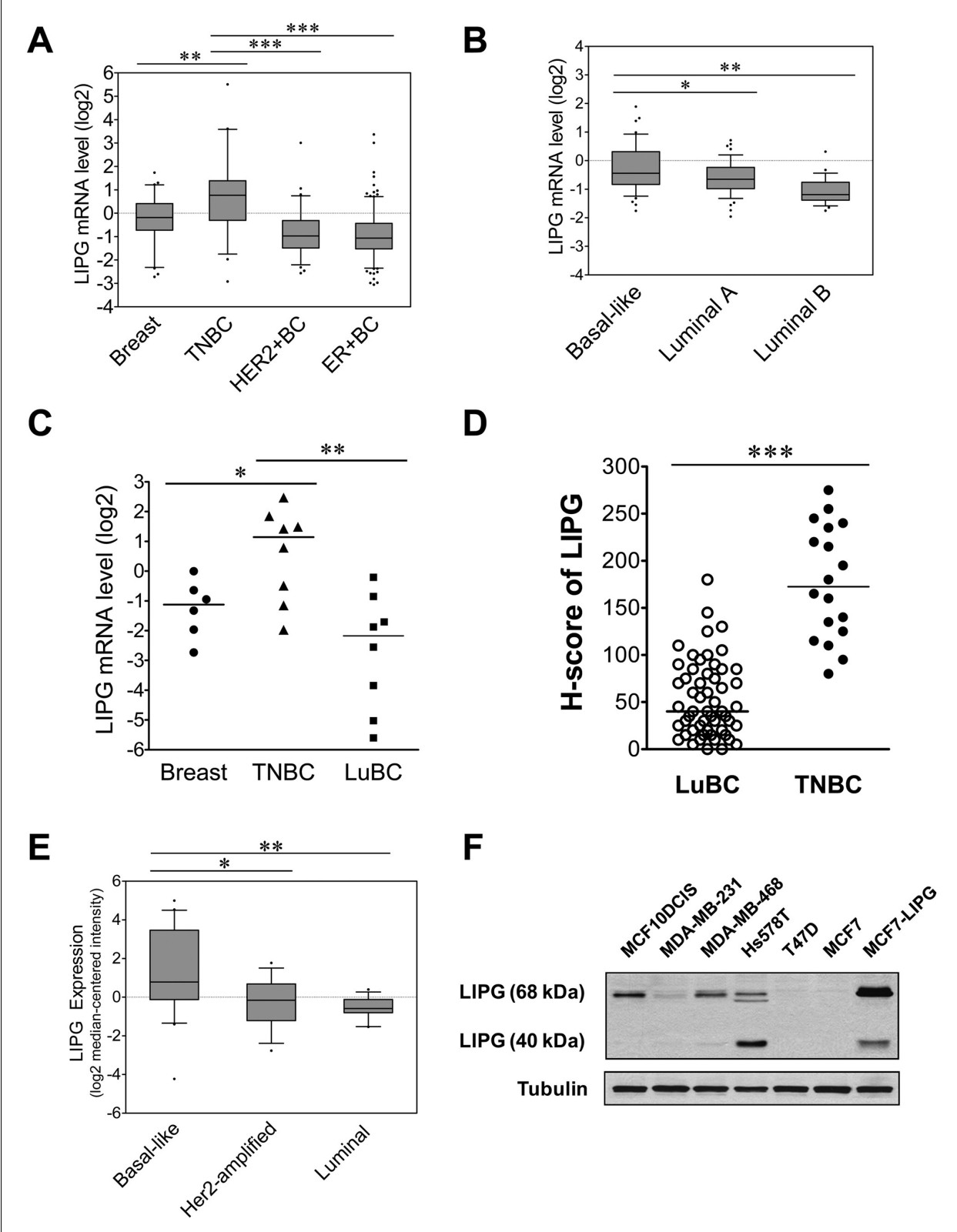

**Figure 1.** LIPG is aberrantly overexpressed in basal-like TNBC. (**A**) LIPG mRNA expression in normal breast and different subtypes of breast cancers based on *in silico* analysis of the TCGA dataset. Normal breast (n = 61), TNBC (n = 46), HER2+ BC (n = 67) and ER+ BC (n = 225). The 25th and 75th percentiles are indicated as a vertical box and the 5th and 95th percentiles are indicated as outliers. (**B**) LIPG mRNA expression in different molecular subtypes of breast cancer classified based on the PAM50 gene expression signature. Expression of LIPG mRNA in basal-like (n = 45), luminal-A (n = 46)

*Figure 1 continued on next page*

*Figure 1 continued*

and luminal-B (n = 25) breast cancers was analyzed *in silico* using the Gluck dataset collected in the Oncomine database. The 25th and 75th percentiles are indicated as a vertical box and the 10th and 90th percentiles are indicated as outliers. (C) qRT-PCR analysis of LIPG mRNA expression in normal breast, TNBC and ER+ PR+ HER2- luminal breast cancers (LuBC). (D) LIPG protein expression in TNBC and LuBC according to H-scoring of IHC-stained tissues using the anti-LIPG antibody. A total of 55 LuBC and 18 TNBC cases were analyzed in the IHC experiment. Horizontal lines indicate medians of IHC datasets. (E) LIPG mRNA expression in different molecular subtypes of breast cancer cell lines. Expression of LIPG mRNA in basal-like (n = 20), Her2-amplified (n = 15) and luminal (n = 13) breast cancer cell lines was analyzed *in silico* using the Hoeflich dataset retrieved from the Oncomine database. The 25th and 75th percentiles are indicated as a vertical box and the 10th and 90th percentiles are indicated as outliers. (F) LIPG protein expression in breast cancer cell lines. MCF7-LIPG is a LIPG-overexpresing MCF7 cell line. Tubulin was used as a loading control. *p<0.05; **p<0.01; ***p<0.001.

DOI: https://doi.org/10.7554/eLife.31334.002

The following source data and figure supplements are available for figure 1:

**Source data 1.** LIPG expression in normal breast and different molecular subtypes of breast cancer.
DOI: https://doi.org/10.7554/eLife.31334.007

**Source data 2.** LIPG expression in different molecular subtypes of breast cancer.
DOI: https://doi.org/10.7554/eLife.31334.008

**Source data 3.** LIPG mRNA and protein expression in LuBC and TNBC.
DOI: https://doi.org/10.7554/eLife.31334.009

**Source data 4.** LIPG expression in different molecular subtypes of breast cancer cell lines.
DOI: https://doi.org/10.7554/eLife.31334.010

**Figure supplement 1.** Expression of LPL is downregulated in breast cancer.
DOI: https://doi.org/10.7554/eLife.31334.003

**Figure supplement 2.** Expression of LIPC is downregulated in breast cancer.
DOI: https://doi.org/10.7554/eLife.31334.004

**Figure supplement 3.** IHC analysis of LIPG protein expression was performed on tissue microarrays of breast cancer.
DOI: https://doi.org/10.7554/eLife.31334.005

**Figure supplement 4.** LIPG protein is overexpressed in TNBC.
DOI: https://doi.org/10.7554/eLife.31334.006

examine the impact of LIPG overexpression on the luminal feature of MCF7 cells, flow cytometric analysis of the luminal marker EpCAM was performed on parental MCF7-CTRL and MCF7-LIPG with or without LIPG knockdown. The FACS data indicated that LIPG overexpression significantly decreased EpCAM$^{high}$ cells (41.92 ± 4.21% of MCF7-LIPG + siControl vs. 61.79 ± 3.11% of MCF7-CTRL + siControl, p<0.01) and LIPG knockdown almost abolished this effect (56.83 ± 2.64%, p<0.01) (*Figure 2E*). To decipher whether LIPG-mediated effects involve changes in expression of basal/EMT and stem cell-related genes, gene expression profiling was performed. The results showed that LIPG overexpression led to increased expression of basal/myoepithelial (*KRT14*), EMT transcription factor (*ZEB1*) and stem cell (*IL6*, *CD133*, *PROCR*, *SOX9*) genes, and decreased expression of E-cadherin (*Figure 2F and G*), whereas LIPG knockdown abolished these expression alterations. These results are consistent with the observed phenotypic changes in MCF7 cells induced by LIPG overexpression. MCF7 cells are known to lack vimentin expression. The immunostaining experiment showed that LIPG overexpression was not sufficient to induce vimentin expression in MCF7 cells (data not shown). As expected, LIPG siRNA-transfected MCF7-CTRL cells showed no significant differences in these phenotypic features when compared to control siRNA-transfected MCF7-CTRL cells (*Figure 2*). These findings together indicate that LIPG functions to promote basal-like features, EMT, and stemness of breast cancer cells.

## LIPG-mediated enhancements of basal/EMT, migratory and cancer stem cell features are largely independent of its lipase catalytic activity

To reveal whether the metabolic role of LIPG is relevant to LIPG-mediated enhancements of basal/EMT, migratory and cancer stem cell features, we created a MCF7 cell line (MCF7-LIPG$^{S149A}$) stably expressing a catalytically inactive mutant of LIPG (S149A), which has been shown to lose the lipase catalytic activity (*Broedl et al., 2003*). By comparing this mutant line with parental and wild-type LIPG-expressing lines, we were able to examine both enzymatic and non-enzymatic effects of LIPG in cancer cells. First, we performed western blot analysis to examine the protein expression level of LIPG$^{S149A}$ in the established MCF7 line. As shown in *Figure 3A*, the MCF7-LIPG$^{S149A}$ line expressed

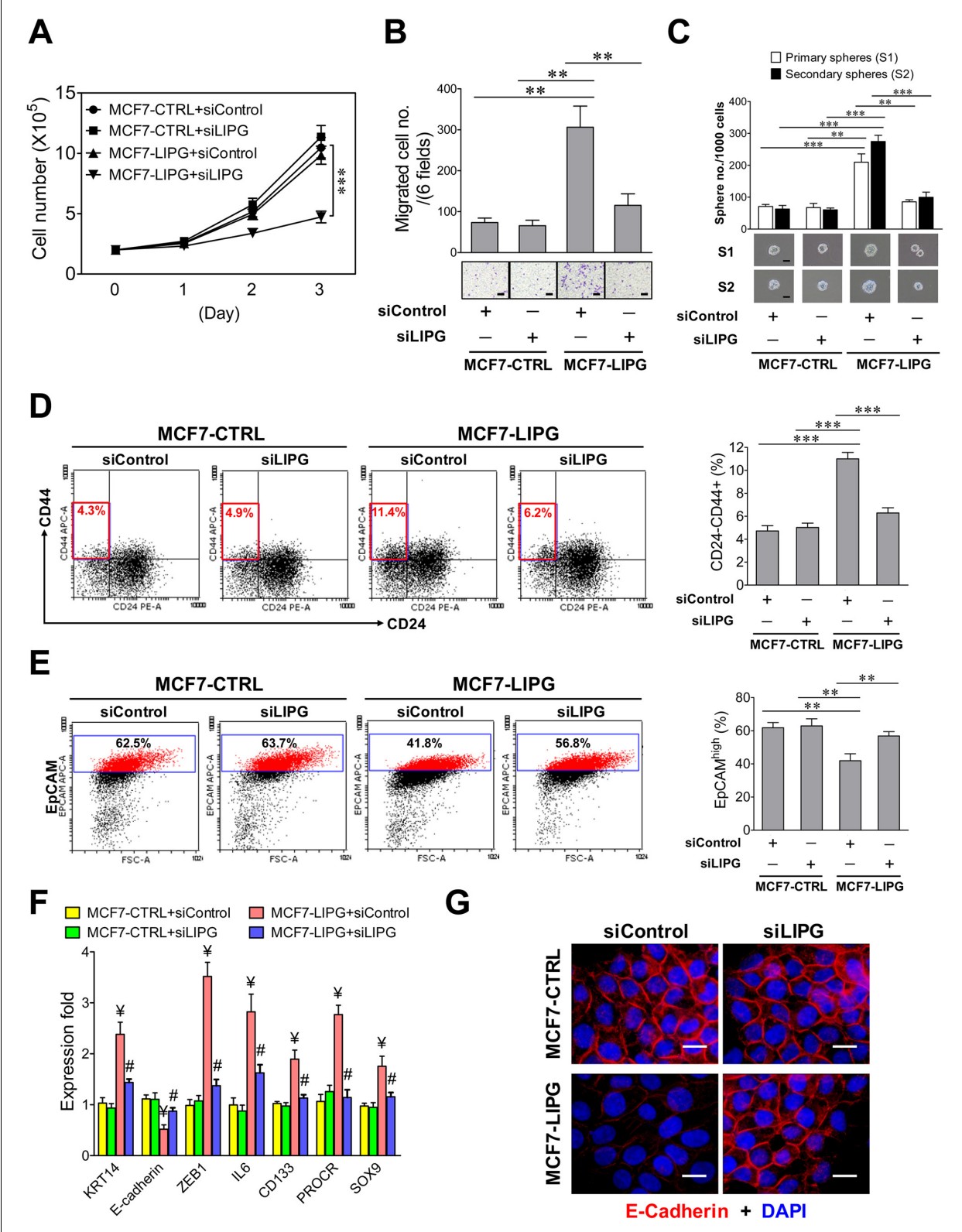

**Figure 2.** Ectopic overexpression of LIPG in LIPG-deficient luminal breast cancer cells promotes migration, stemness and basal/EMT features. (A) Growth of LIPG-overexpressing MCF7 cells, but not parental control cells, is inhibited by LIPG knockdown. $2 \times 10^5$ cells of MCF7-CTRL (the vector-control cells) and MCF7-LIPG (LIPG-overexpressing cells) transfected with either the control siRNA (siControl) or LIPG siRNA (siLIPG) were seeded for the 3-day cell growth study (n = 3 for each time point). (B) LIPG overexpression promotes migration of MCF7 cells. Triplicate transwell-based migration

*Figure 2 continued on next page*

*Figure 2 continued*

experiments were performed on siRNA-transfected cells as described in (**A**). The scale bar indicates 100 µm. (**C**) LIPG overexpression enhances the primary and secondary CSC sphere formation of MCF7. Triplicate primary sphere formation experiments were performed on siRNA-transfected cells as described in (**A**). After a week, primary spheres were counted and collected for siRNA transfections and secondary sphere formation analysis as described in 'Materials and methods'. The scale bar indicates 50 µm. (**D**) LIPG overexpression increases the CSC-enriched CD24-/CD44+ cell subset in MCF7 cells. Triplicate FACS analyses of the CD24/CD44 profile were performed on siRNA-transfected cells as described in (**A**). The representative FACS profiles of CD24/CD44 are shown on the left panel and the bar graph is shown on the right panel. (**E**) LIPG overexpression decreases the EpCAM^high cell population in MCF7 cells. Triplicate FACS analyses of EpCAM vs. FSC (Forward Scatter parameter) were performed on siRNA-transfected cells as described in (**A**). The representative FACS profiles of EpCAM/FSC are shown on the left panel and the bar graph is shown on the right panel. (**F**) LIPG overexpression in MCF7 cells leads to the increased expression of basal/EMT and stem cell genes and downregulation of E-cadherin. qRT-PCR analysis of basal/EMT and stem cell genes was performed on siRNA-transfected cells as described in (**A**). ¥ p<0.01 versus the control datasets (MCF7-CTRL + siControl and MCF7-CTRL + siLIPG); # p<0.05 versus the dataset of MCF7-LIPG cells transfected with siControl (MCF7-LIPG + siControl); n = 3. (**G**) LIPG overexpression downregulates E-cadherin expression. Immunofluorescent analysis of E-cadherin protein expression was performed on siRNA-transfected cells as described in (**A**). Cellular DNA was stained with DAPI. The scale bar indicates 25 µm. Error, standard deviation (SD); **p<0.01; ***p<0.001.

DOI: https://doi.org/10.7554/eLife.31334.011

The following source data and figure supplement are available for figure 2:

**Source data 1.** Source data for *Figure 2.*
DOI: https://doi.org/10.7554/eLife.31334.013
**Figure supplement 1.** Western blot analysis of LIPG and tubulin protein expression in MCF7-CTRL, MCF7-LIPG and MCF7-LIPG plus LIPG knockdown.
DOI: https://doi.org/10.7554/eLife.31334.012

the LIPG mutant protein at the level similar to that of the wild-type LIPG-expressing MCF7 line (MCF7-LIPG). We next examined the growth rate of MCF7-LIPG^S149A cells for 4 days by comparison with the parental control (MCF7-CTRL) and MCF7-LIPG cells. We found that LIPG^S149A overexpression inhibited approximately 50% of the MCF7 cell growth rate, while wild-type LIPG overexpression only suppressed approximately 20% of the MCF7 cell growth rate (*Figure 3B*). The absence of the lipase catalytic activity of LIPG in MCF7-LIPG^S149A cells led to loss of approximate 30% of the cell growth rate when compared to MCF7-LIPG cells, suggesting that the lipase activity of LIPG plays a role in promoting cell growth.

To determine the impact of LIPG^S149A overexpression on cell migration, we performed transwell-based migration assays. Interestingly, overexpression of LIPG^S149A was highly effective to induce an approximately seven-fold increase in migration of MCF7 cells, which was stronger than the five-fold increase caused by wild-type LIPG overexpression (p<0.05) (*Figure 3C*). This finding indicates that the lipase function of LIPG is not required for LIPG to promote MCF7 migration. With regard to the impact of LIPG^S149A overexpression on CSCs, we conducted primary and secondary CSC sphere formation assays. As shown in *Figure 3D*, LIPG^S149A was competent to enhance primary and secondary CSC sphere formation of MCF7 in spite of the fact that CSC spheres were smaller compared with those from control MCF7-CTRL cells. The extent of this CSC enhancement was similar to that of MCF7-LIPG. Nevertheless, wild-type LIPG overexpression led to the formation of larger CSC spheres, which was in contrast to smaller CSC spheres formed from MCF7-LIPG^S149A (*Figure 3D*). These findings suggest that although the effect of LIPG on the sphere formation (self-renewal) ability of CSCs is largely independent of its lipase activity, this metabolic lipase function is required for LIPG to promote the CSC proliferation rate, resulting in increased sphere sizes. Moreover, flow cytometric analysis showed that LIPG^S149A was proficient in increasing the CD24-/CD44+ CSC population in MCF7 cells and its increased extent was greater than that of MCF7-LIPG (15.06 ± 0.75% of MCF7-LIPG^S149A vs. 10.77 ± 1.19% of MCF7-LIPG, p<0.01) (*Figure 3E*). We noted that the sphere data of MCF7-LIPG^S149A (197 ± 28 primary CSC spheres/1000 cells from MCF7-LIPG^S149A vs. 242 ± 18 primary CSC spheres/1000 cells from MCF7-LIPG, p=0.0806) (*Figure 3D*) did not reflect the further increased CD24-/CD44+ population observed in MCF7-LIPG^S149A (*Figure 3E*). Given that the sphere result is the combined outcome of both self-renewal and the proliferation rate of CSCs, some of the formed CSC spheres from MCF7-LIPG^S149A were too small (<50 µm) to be counted due to low sphere cell proliferation and they were excluded from the sphere count. This caused the discrepancy between the results of CSC sphere and CD24/CD44 profiling analyses.

To determine the impact of LIPG^S149A on basal/EMT features of breast cancer cells, we examined expression of basal/EMT genes in MCF7- LIPG^S149A compared with MCF7-CTRL and MCF7-LIPG

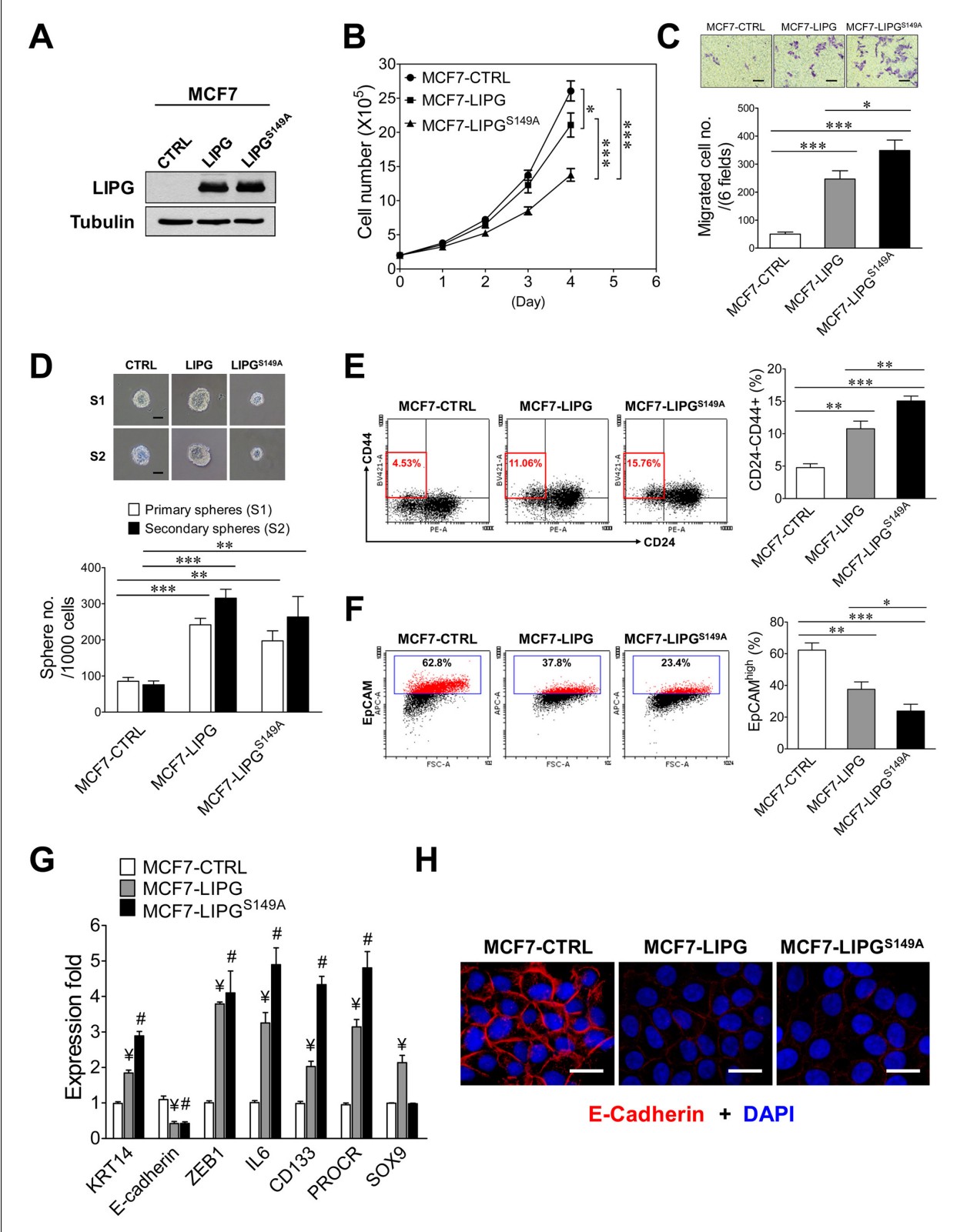

**Figure 3.** The lipase catalytic activity of LIPG is relevant to cell growth, but not to the motility, stemness and EMT of breast cancer cells. (A) Western blot analysis of LIPG and tubulin protein expression in vector-control (CTRL), wild-type LIPG-overexpressing (LIPG) and LIPG mutant-overexpressing (LIPG$^{S149A}$) MCF7 lines. (B) Loss of the lipase catalytic activity of LIPG attenuates cell growth. $2 \times 10^5$ cells of three MCF7 lines as described in (A) were seeded for the four-day cell growth study (n = 3 for each time point). (C) LIPG$^{S149A}$ is potent to promote migration of MCF7 cells. Triplicate transwell-

*Figure 3 continued on next page*

*Figure 3 continued*

based migration experiments were performed on three MCF7 lines as described in (**A**). The scale bar indicates 100 μm. (**D**) LIPG$^{S149A}$ is competent to enhance the primary and secondary CSC sphere formation of MCF7 but fails to promote sphere cell proliferation. Triplicate primary and secondary sphere formation experiments were performed on three MCF7 lines as described in (**A**). The scale bar indicates 50 μm. (**E**) LIPG$^{S149A}$ overexpression increases the CSC-enriched CD24-/CD44+ cell subset in MCF7 cells. Triplicate FACS analyses of the CD24/CD44 profile were performed on three MCF7 lines as described in (**A**). The representative FACS profiles of CD24/CD44 are shown on the left panel and the bar graph is shown on the right panel. (**F**) LIPG$^{S149A}$ overexpression decreases the EpCAM$^{high}$ cell population in MCF7 cells. Triplicate FACS analyses of EpCAM vs. FSC were performed on three MCF7 lines as described in (**A**). The representative FACS profiles of EpCAM/FSC are shown on the left panel and the bar graph is shown on the right panel. (**G**) LIPG$^{S149A}$ overexpression in MCF7 cells leads to the increased expression of basal/EMT and stem cell genes and downregulation of E-cadherin. qRT-PCR analysis of basal/EMT and stem cell genes was performed on three MCF7 lines as described in (**A**). ¥ p<0.01 versus the control dataset (MCF7-CTRL); # p<0.01 versus the control dataset (MCF7-CTRL); n = 3. (**H**) LIPG$^{S149A}$ overexpression downregulates E-cadherin expression. Immunofluorescent analysis of E-cadherin protein expression was performed on three MCF7 lines as described in (**A**). Cellular DNA was stained with DAPI. The scale bar indicates 25 μm. Error, standard deviation (SD); *p<0.05; **p<0.01; ***p<0.001.

DOI: https://doi.org/10.7554/eLife.31334.014

The following source data is available for figure 3:

**Source data 1.** Source data for *Figure 3.*

DOI: https://doi.org/10.7554/eLife.31334.015

using flow cytometry, qRT-PCR and immunofluorescence analyses. We found that similar to wild-type LIPG, LIPG$^{S149A}$ was capable of downregulating expression of EpCAM and E-cadherin and upregulating expression of *KRT14* and *ZEB1* genes (***Figure 3F, G and H***). These data suggest that the lipase activity of LIPG is dispensable for the LIPG-mediated promotion of basal/EMT phenotypes in breast cancer cells, consistent with the migration data (***Figure 3C***). In line with CSC sphere and CD24/CD44 data, LIPG$^{S149A}$ had most of the wild-type LIPG ability to enhance expression of stem cell genes (*IL6*, *CD133* and *PROCR*) (***Figure 3G***). These findings, taken together, suggest that LIPG possesses both lipase-dependent and lipase-independent functions in promoting various tumorigenic characteristics of breast cancer cells.

## LIPG is required for the tumorigenicity and metastasis of basal-like TNBC

To reveal the role of LIPG in basal-like TNBC, we performed LIPG knockdown studies on basal-like TNBC cell lines MCF10DCIS and MDA-MB-468, both of which express significant levels of full-length 68 kDa LIPG protein (***Figure 1F***). The knockdown efficiency of two different siRNAs was confirmed by western blot analysis (***Figure 4—figure supplement 1***). We found that LIPG knockdown in MCF10DCIS by either siRNA markedly suppressed cell growth, migration, invasion and CSC sphere formation (***Figure 4A, B, C and D***). These alterations correlated with increased EpCAM-positive cells, downregulated expression of multiple stem-cell and basal/EMT programming genes, elevated E-cadherin protein levels and decreased vimentin protein levels (***Figure 4E, F, G, H and I***). These findings indicate that LIPG expression is required for tumorigenic, basal-like and EMT characteristics of MCF10DCIS cells.

In order to study the role of LIPG in tumorigenicity and metastasis of basal-like TNBC, we created two stable LIPG-knockdown MDA-MB-468 cell lines, each of which expressed a distinct LIPG shRNA. LIPG knockdown of these two lines was confirmed by western blot analysis (***Figure 5—figure supplement 1***). Consistent with the MCF10DCIS results, LIPG deficiency substantially impaired cell growth, migration, invasion and CSC sphere formation of MDA-MB-468 cells (***Figure 5A, B, C and D***). Xenograft studies also demonstrated that LIPG depletion completely inhibited the *in vivo* tumor development of MDA-MB-468 in nude mice (***Figure 5E***). Consistently, LIPG inactivation resulted in downregulating expression of multiple stem-cell and EMT programming genes, including downregulation of vimentin protein expression (***Figure 5F and G*** and ***Figure 5—figure supplement 2***). MDA-MB-468 is an E-cadherin-negative cell line. The immunostaining experiment showed that LIPG knockdown was not sufficient to induce E-cadherin expression in MDA-MB-468 cells (data not shown). Moreover, over 50% of LIPG-depleted MDA-MB-468 cells expressed the luminal protein marker EpCAM compared to only 3% of the control cells showing EpCAM positivity (***Figure 5H***), indicating that loss of LIPG significantly promotes the luminal feature in MDA-MB-468 cells. This finding correlated with a morphological change in LIPG-knockdown MDA-MB-468 cell lines that manifested

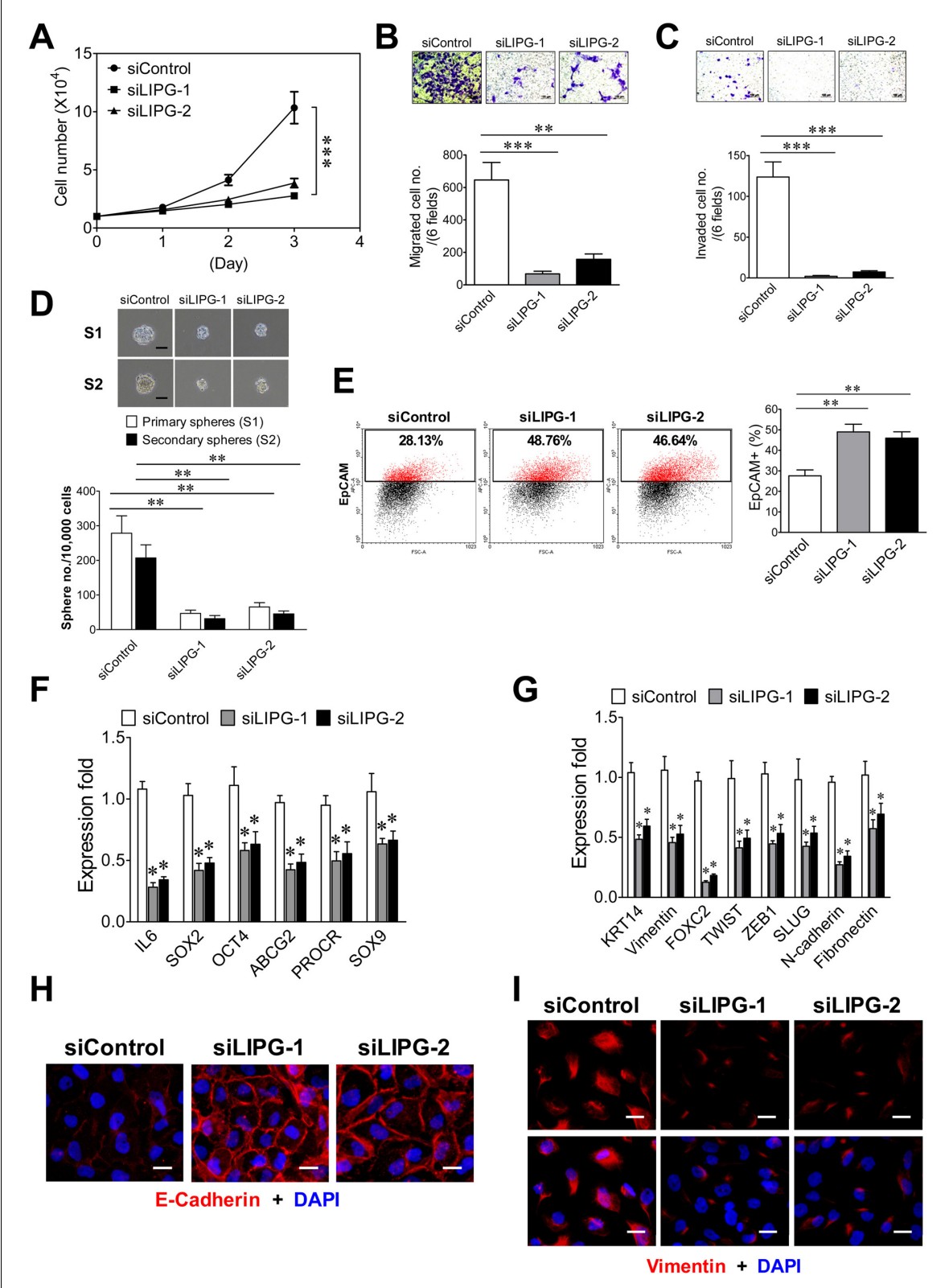

**Figure 4.** LIPG inactivation impairs invasiveness, stemness and basal/EMT features of basal-like DCIS cells. (**A**) LIPG knockdown inhibits cell growth of MCF10DCIS cells. $1 \times 10^4$ cells of siRNA-transfected cells were seeded for the three-day cell growth study. (**B**) LIPG knockdown impairs migration of MCF10DCIS cells. The scale bar indicates 100 μm. (**C**) LIPG knockdown suppresses invasion of MCF10DCIS cells. The scale bar indicates 100 μm. (**D**) LIPG knockdown leads to decreases in primary and secondary sphere formation of MCF10DCIS cells. The scale bar indicates 50 μm. (**E**) LIPG

*Figure 4 continued on next page*

*Figure 4 continued*

knockdown increases the EpCAM-positive cell population in MCF10DCIS cells. (F) LIPG knockdown results in the downregulation of stem cell gene expression in MCF10DCIS cells. (G) LIPG knockdown leads to the downregulation of basal/EMT gene expression. (H) LIPG knockdown causes increased E-cadherin expression in MCF10DCIS cells. Immunofluorescent analysis of E-cadherin was performed on siRNA-transfected MCF10DCIS cells. Cellular DNA was stained with DAPI. The scale bar indicates 25 µm. (I) LIPG knockdown gives rise to decreased vimentin expression in MCF10DCIS cells. Immunofluorescent analysis of vimentin was performed on siRNA-transfected MCF10DCIS cells. Cellular DNA was stained with DAPI. The scale bar indicates 25 µm. Error, SD (n = 3); *p<0.05; **p<0.01; ***p<0.001.

DOI: https://doi.org/10.7554/eLife.31334.016

The following source data and figure supplement are available for figure 4:

**Source data 1.** The source data file contains numerical data that were used to generate data graphs presented in sub-figures (4A, B, C, D, E, F, G) of *Figure 4.*
DOI: https://doi.org/10.7554/eLife.31334.018

**Figure supplement 1.** Western blot analysis of LIPG and tubulin protein expression in MCF10DCIS cells with or without LIPG knockdown by two different siRNAs.
DOI: https://doi.org/10.7554/eLife.31334.017

epithelial-like morphology in contrast to the mesenchymal-like morphology of the control cell line (*Figure 5—figure supplement 3*). To determine the role of LIPG in TNBC metastasis *in vivo*, we conducted animal metastasis assays. As shown in *Figure 5I*, found tumor foci were identified in the animal group injected with control scramble shRNA-expressing MDA-MB-468 cells, but not in the group injected with LIPG shRNA-expressing cells. These tumor foci formed in the lungs of the control group exhibited strong IHC staining of vimentin in contrast to vimentin negativity in lung tissue. As MDA-MB-468 cells expressed the high levels of vimentin (*Figure 5—figure supplement 2*), IHC results have confirmed that these tumor foci found in lungs were derived from metastasis of MDA-MB-468. Statistically analyzing numbers of lung metastatic tumor foci in both animal groups showed that LIPG knockdown by shRNA completely abrogated *in vivo* lung metastasis of MDA-MB-468 cells in nude mice (p<0.01). These findings demonstrate that LIPG is crucial for malignancies and metastasis of basal-like TNBC.

## ISG15 mediates the oncogenic effect of LIPG in basal-like TNBC cells

To unravel the underlying mechanism mediating the oncogenic function of LIPG, we performed proteomic profiling analysis on MCF7-LIPG and parental control cells. Surprisingly, among upregulated proteins (the fold ≥2.0) we identified in MCF7-LIPG cells, 15 out of them are encoded by interferon-stimulated genes (ISGs). Eight ISGs (*B2M*, *IFITM2*, *ISG15*, *MVP*, *NEDD4*, *S100A8*, *S100A9*, *UBE2L6*) belong to IFN type I and II, five ISGs (*AGR2*, *EIF2B2*, *JARID2*, *S100A7*, *TTC21A*) are specific to IFN type I, and two ISGs (*AKR1C2*, *CTSD*) are specific to IFN type II (*Figure 6—figure supplement 1*). Among these upregulated ISG proteins, ISG15 and UBE2L6/UBCH8 were of interest. ISG15 has been shown to be involved in some subtypes of breast cancer, including TNBC (*Hadjivasiliou, 2012*; *Burks et al., 2014*; *Forys et al., 2014*), and UBE2L6/UBCH8, a ubiquitin conjugating E2 enzyme, has been reported to mediate ISG15 conjugation (*Zhao et al., 2004*). Therefore, we hypothesized that ISG15 mediates the oncogenic function of LIPG. To test this hypothesis, we first examined whether ISG15 expression is LIPG-dependent. From qRT-PCR and western blot analyses, we found that knockdown of endogenous LIPG in MCF10DCIS and MDA-MB-468 downregulated the mRNA and protein expression of ISG15 (*Figure 6A and B*). Furthermore, ectopic expression of LIPG in MCF7, MCF10DCIS and MDA-MB-468 led to the induction of ISG15 mRNA and protein expression, which was abolished by concurrent LIPG siRNA knockdown (*Figure 6A and B*). We noticed that a basal level of ISG15 protein expression was detectable in parental MCF7 cells (*Figure 6B*), suggesting that there is a LIPG-independent mechanism for sustaining ISG15 expression in LIPG-deficient luminal breast cancer cells. Interestingly, ISG15 expression became reliant on LIPG in MCF7-LIPG cells, suggesting that LIPG overexpression might have reprogrammed MCF7 cells and rendered ISG15 expression dependent on LIPG. Moreover, the catalytically inactive mutant of LIPG (LIPG$^{S149A}$) was capable of upregulating ISG15 expression in MCF7 cells (*Figure 6C*), suggesting that the lipase activity of LIPG is dispensable for the LIPG-ISG15 signaling axis. Relevantly, this positive regulatory relationship between LIPG and ISG15 was further confirmed by *in silico* analysis of 45 cases of BLBC using the Oncomine database (*Figure 6D*).

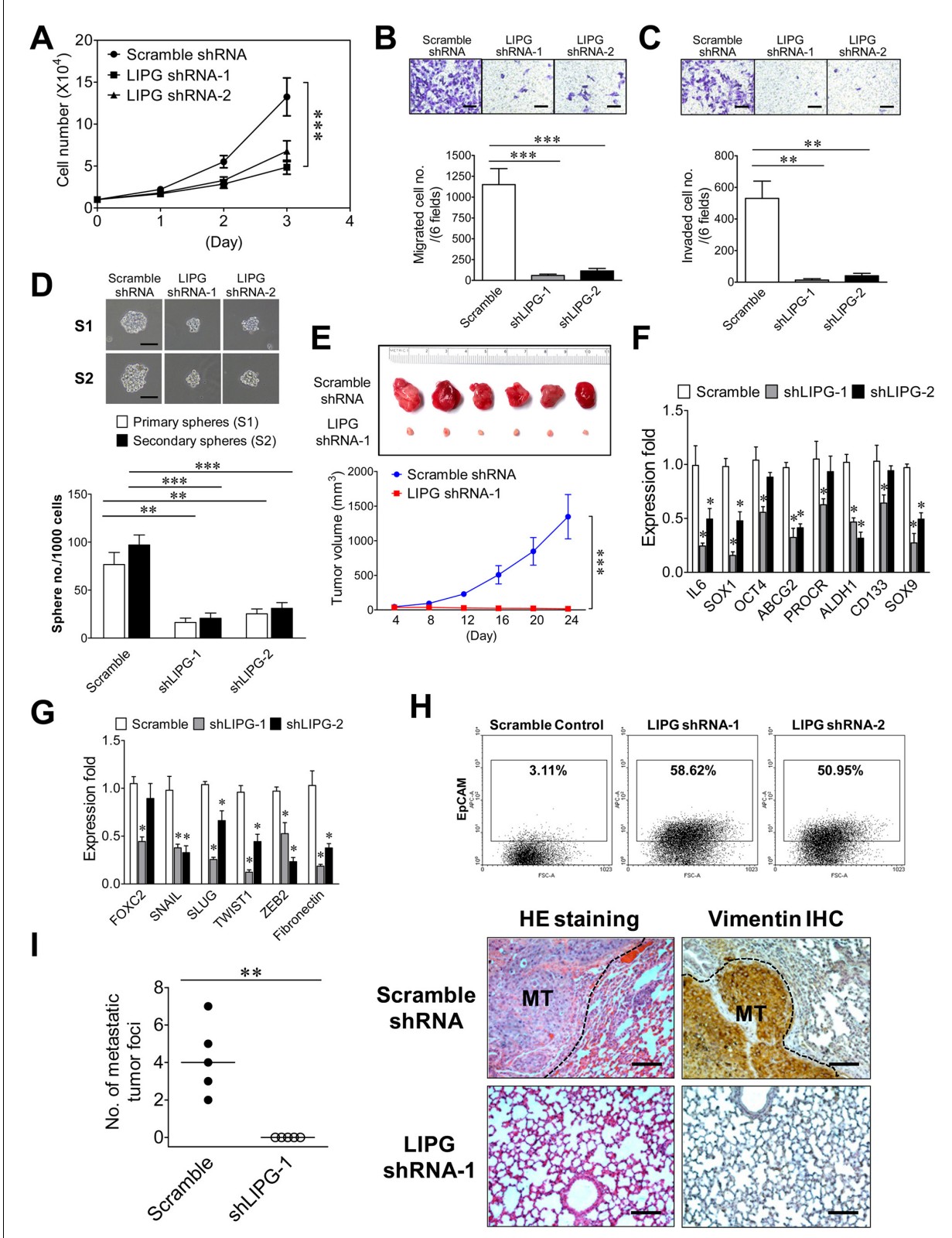

**Figure 5.** LIPG is required for tumorigenicity and metastasis of basal-like TNBC cells. (A) LIPG inactivation attenuates *in vitro* cell growth of MDA-MB-468. $1 \times 10^4$ cells of shRNA-expressing cells were seeded for the 3-day cell growth study. (B) LIPG depletion impairs migration of MDA-MB-468 cells. The scale bar indicates 100 μm. (C) LIPG deficiency inhibits invasion of MDA-MB-468 cells. The scale bar indicates 100 μm. (D) LIPG knockdown suppresses primary and secondary CSC sphere formation of MDA-MB-468. The scale bar indicates 50 μm. (E) Loss of LIPG abrogates *in vivo* tumor

*Figure 5 continued on next page*

*Figure 5 continued*

formation of MDA-MB-468. 5 × 10⁶ of scramble or LIPG shRNA-expressing MDA-MB-468 cells were transplanted into the mammary fat pads of nude mice for xenograft tumor formation (n = 6 for each experimental group). The picture of isolated tumors and plotted tumor growth curves are shown on the top and bottom panels, respectively. (**F**) LIPG knockdown downregulates the expression of stem cell genes in MDA-MB-468 cells. (**G**) LIPG knockdown downregulates the expression of EMT programming genes in MDA-MB-468 cells. (**H**) LIPG inactivation substantially increases EpCAM+ cells in MDA-MB-468. (**I**) LIPG inactivation abrogates *in vivo* lung metastasis of MDA-MB-468 cells. Tail vein injection of either scramble shRNA-expressing or LIPG shRNA-expressing MDA-MB-468 cells (1 × 10⁶) was performed on nude mice (n = 5 for each experimental group). After 4 weeks, mice were euthanized and dissected to isolate their lungs for HE staining and IHC analysis of vimentin. In contrast to the negativity of vimentin staining in lung tissue, the strong vimentin staining in tumor tissue areas has confirmed that these formed tumor foci were derived from metastasis of MDA-MB-468, a TNBC line expressing the high levels of vimentin according to the immunofluorescence study. Metastatic tumor foci were counted according to HE and IHC staining results. Representative HE and IHC staining pictures are shown on the right panel and the statistical analysis of metastatic tumor foci is shown on the left panel. The medians (indicated by horizontal lines) for both experimental groups are shown in the plot. The metastatic tumor (MT) tissue areas in the lung HE and IHC staining pictures are indicated by the black dash line. The scale bar indicates 100 µm. Error, SD (n = 3); *p<0.05; **p<0.01; ***p<0.001.

DOI: https://doi.org/10.7554/eLife.31334.019

The following source data and figure supplements are available for figure 5:

**Source data 1.** Source data for *Figure 5.*

DOI: https://doi.org/10.7554/eLife.31334.023

**Figure supplement 1.** Western blot analysis of LIPG and tubulin protein expression in MDA-MB-468 cells with or without LIPG knockdown by two different shRNAs.

DOI: https://doi.org/10.7554/eLife.31334.020

**Figure supplement 2.** Immunofluorescent analysis of vimentin protein expression was performed on MDA-MB-468 lines expressing either the scramble shRNA or two different LIPG shRNAs.

DOI: https://doi.org/10.7554/eLife.31334.021

**Figure supplement 3.** LIPG knockdown by two different shRNAs induced a morphological change in MDA-MB-468 cells from a mesenchymal-like phenotype to an epithelial-like phenotype.

DOI: https://doi.org/10.7554/eLife.31334.022

To reveal the functional role of ISG15 in basal-like TNBC, we performed ISG15 knockdown on MCF10DCIS and MDA-MB-468 cells using a validated siRNA that has been reported elsewhere (*Zhang et al., 2015a*). ISG15 knockdown in both cell lines by this siRNA was confirmed by western blot analysis (*Figure 6—figure supplement 2*). We observed that ISG15 knockdown had no impact on LIPG protein levels (*Figure 6—figure supplement 2*), indicating that there is no feedback regulation occurring in the LIPG-ISG15 axis. The knockdown studies showed that loss of ISG15 moderately impaired cell growth and CSC sphere formation of MCF10DCIS and MDA-MB-468, but substantially inhibited migration and invasion of both cell lines (*Figure 6E, F, G and H*). These data indicate that the invasiveness of MCF10DCIS and MDA-MB-468 highly relies on ISG15 function. Consistently, ISG15-knockdown MCF10DCIS cells manifested a significantly decreased basal feature, indicated by a dramatic increase in the percentage of EpCAM+ cells in ISG15-knockdown MCF10DCIS (from 31.93 ± 6.53% of the control to 60.66 ± 5.78%, p<0.01) (*Figure 6I*). Moreover, qRT-PCR studies showed that ISG15 depletion in MCF10DCIS cells profoundly downregulated expression of multiple EMT programming genes (*Figure 6J*). In line with the qRT-PCR data, immunofluorescent analysis showed that ISG15 knockdown led to a reduction in EMT features of TNBC cells, characterized by E-cadherin upregulation in MCF10DCIS and vimentin downregulation in both MCF10DCIS and MDA-MB-468 (*Figure 6K* and *Figure 6—figure supplement 3*). In contrast, only two stem cell genes were affected by ISG15 knockdown in MCF10DCIS cells (*Figure 6J*). These data suggest that ISG15 more selectively regulates EMT genes than stem-cell genes, consistent with the observed functional effects of ISG15. These findings together imply that ISG15 mainly mediates the basal/EMT-promoting function of LIPG.

To demonstrate that ISG15 functions downstream of LIPG and mediates the LIPG effect, we examined the role of ISG15 in LIPG-overexpressing MDA-MB-468 cells that showed an approximately two-fold increase in invasion (*Figure 6L*). Similar to LIPG knockdown, ISG15 knockdown inhibited approximately 90% invasion of LIPG-overexpressing MDA-MB-468 cells (*Figure 6L*). Consistently, ISG15 knockdown abolished the increased expression of three EMT programming genes in LIPG-overexpressing MDA-MB-468 cells, similar to the effect of LIPG knockdown (*Figure 6M*). These

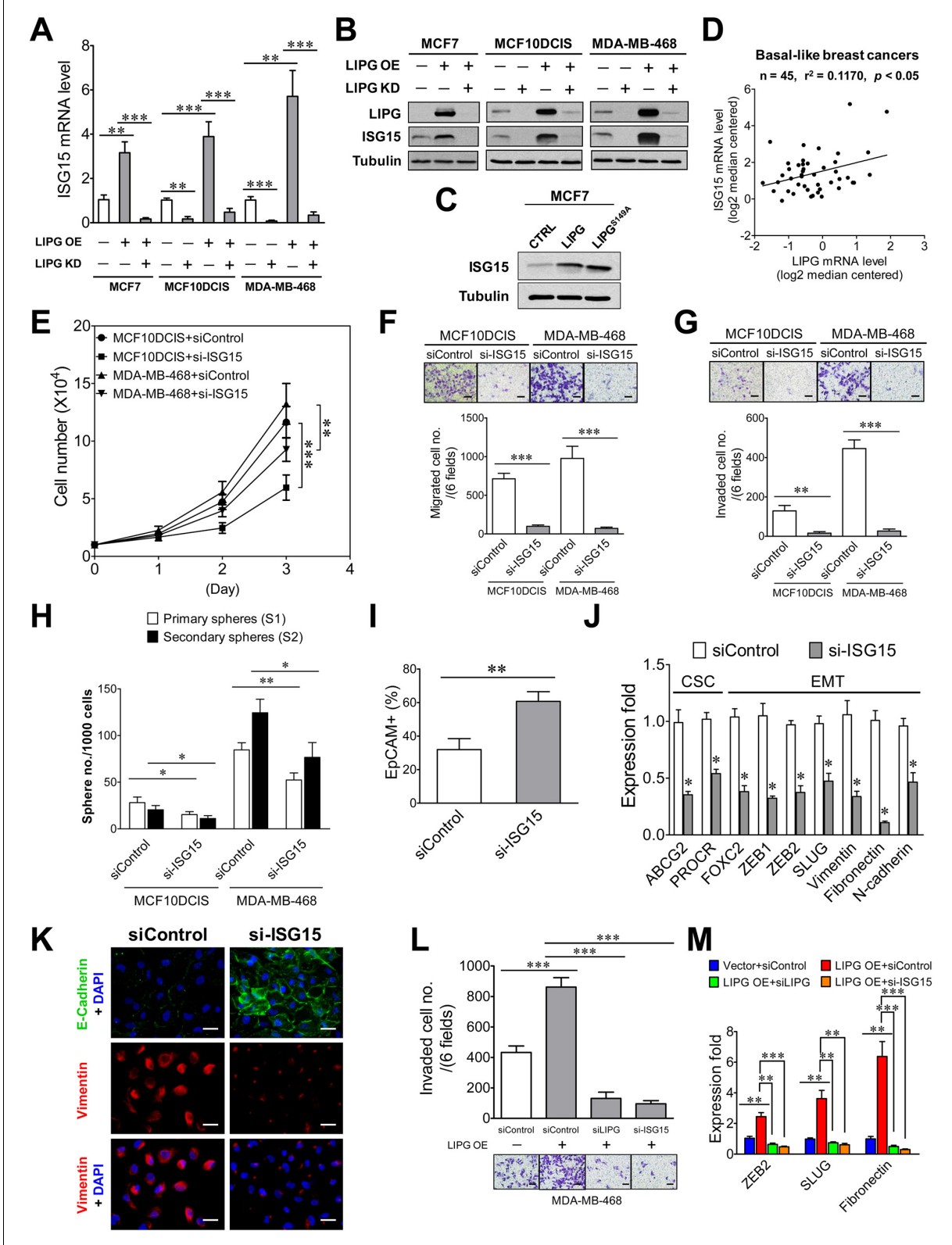

**Figure 6.** ISG15 is the downstream mediator of oncogenic LIPG signaling in basal-like TNBC cells. (**A**) ISG15 mRNA expression is LIPG-dependent. qRT-PCR analysis of ISG15 mRNA expression was performed on indicated cells with or without LIPG overexpression (LIPG OE) in combination with or without LIPG knockdown (LIPG KD). (**B**) ISG15 protein expression is LIPG-dependent. Western blot analysis of LIPG, ISG15 and tubulin was performed on protein lysates isolated from cell samples as described in (**A**). (**C**) The lipase catalytic activity of LIPG is dispensable for LIPG-dependent expression

*Figure 6 continued on next page*

*Figure 6 continued*

of ISG15. Western blot analysis of ISG15 and tubulin was performed on vector-control, wild-type LIPG-overexpressing and LIPG$^{S149A}$-overexpressing MCF7 cells. (D) ISG15 mRNA expression is positively correlated with LIPG expression in human basal-like breast cancers. The linear regression correlation was plotted based on *in silico* analysis of ISG15 and LIPG mRNA levels in 45 cases of BLBC derived from the Gluck dataset collected in the Oncomine database. (E) ISG15 knockdown impairs cell growth of MCF10DCIS and MDA-MB-468. (F) ISG15 knockdown inhibits migration of MCF10DCIS and MDA-MB-468 cells. The scale bar indicates 100 µm. (G) ISG15 knockdown suppresses invasion of MCF10DCIS and MDA-MB-468. The scale bar indicates 100 µm. (H) ISG15 knockdown attenuates primary and secondary CSC sphere formation of MCF10DCIS and MDA-MB-468. (I) ISG15 knockdown increases EpCAM+ cells in MCF10DCIS. (J) ISG15 knockdown downregulates the expression of stem cell and EMT genes in MCF10DCIS cells. (K) ISG15 knockdown leads to E-cadherin upregulation and vimentin downregulation in MCF10DCIS cells. The scale bar indicates 25 µm. (L) ISG15 is required for LIPG overexpression-promoted invasion of MDA-MB-468. Transwell-based invasion analysis was performed on parental control MDA-MB-468 cells and LIPG-overexpressing cells with or without knockdown of either LIPG or ISG15. The scale bar indicates 100 µm. (M) ISG15 is required for LIPG-overexpression-enhanced expression of EMT genes in MDA-MB-468 cells. qRT-PCR analysis of three EMT genes was performed on mRNA samples isolated from cells as described in (L). Error, SD (n = 3); *p<0.05; **p<0.01; ***p<0.001.

DOI: https://doi.org/10.7554/eLife.31334.024

The following source data and figure supplements are available for figure 6:

**Source data 1.** ISG15 mRNA expression is LIPG-dependent.

DOI: https://doi.org/10.7554/eLife.31334.028

**Source data 2.** Source data for *Figure 6*.

DOI: https://doi.org/10.7554/eLife.31334.029

**Figure supplement 1.** LIPG overexpression in luminal MCF7 breast cancer cells leads to increased expression of interferon-stimulated genes (ISGs).

DOI: https://doi.org/10.7554/eLife.31334.025

**Figure supplement 2.** Western blot analysis of ISG15, LIPG and tubulin protein expression in MCF10DCIS and MDA-MB468 cells with or without ISG15 knockdown.

DOI: https://doi.org/10.7554/eLife.31334.026

**Figure supplement 3.** Immunofluorescent analysis of vimentin protein expression was performed on either control siRNA-(siControl)-transfected or ISG15 siRNA (si-ISG15)-transfected MDA-MB-468 cells.

DOI: https://doi.org/10.7554/eLife.31334.027

findings demonstrate that ISG15 is the downstream mediator of LIPG signaling and mainly functions as a promoter of invasiveness and the basal phenotype.

## LIPG is involved in the oncogenic DTX3L-ISG15 signaling axis

We next sought to identify what molecular regulators are involved in aberrant overexpression of LIPG in basal-like TNBC. Given that LIPG overexpression in MCF7 cells induced expression of IFN signature genes, including ISG15, we tested whether IFN signaling regulators are involved in modulating LIPG expression in TNBC cells. Among these known regulators, DTX3L drew our interest as it activates expression of ISG15 and other IFN-responsive genes by regulating the mono-ubiquitination of histones and has been found to function as an oncogenic factor in metastasis of melanoma (*Thang et al., 2015*; *Zhang et al., 2015b*). Due to these potential facts and the unknown function of DTX3L in breast cancer, we performed DTX3L knockdown studies to reveal its role in LIPG signaling in basal-like TNBC cells.

Western blot studies showed that two different DTX3L siRNAs successfully knocked down over 90% of DTX3L protein expression in MCF10DCIS and MDA-MB-468 cells (*Figure 7A*). Surprisingly, DTX3L knockdown by either siRNA resulted in dramatic downregulation of LIPG protein expression in both cell lines (*Figure 7A*). To determine whether the DTX3L-dependent expression of LIPG is regulated at the mRNA level, we performed qRT-PCR experiments on these siRNA-transfected cell samples. As shown in *Figure 7B*, DTX3L knockdown only led to approximately 25% and 40% decreases in LIPG mRNA levels of MCF10DCIS and MDA-MB-468 cells, respectively. Since the modest decrease of LIPG mRNA levels caused by DTX3L deficiency could not completely account for the dramatic change in LIPG protein levels, we next tested whether DTX3L regulates LIPG protein stability. We examined LIPG protein levels in control and DTX3L-knockdown MDA-MB-468 cells treated with the protein synthesis inhibitor cycloheximide in a time course experiment. The result showed that DTX3L-knockdown cells displayed a shorter LIPG protein half-life (approximate 1 hr) compared to control cells with a longer LIPG half-life (approximate 4 hr) (*Figure 7C*), indicating that loss of DTX3L significantly decreased LIPG protein stability. To decipher whether this observed event was attributable to enhanced degradation of LIPG protein, we treated DTX3L siRNA-transfected cells

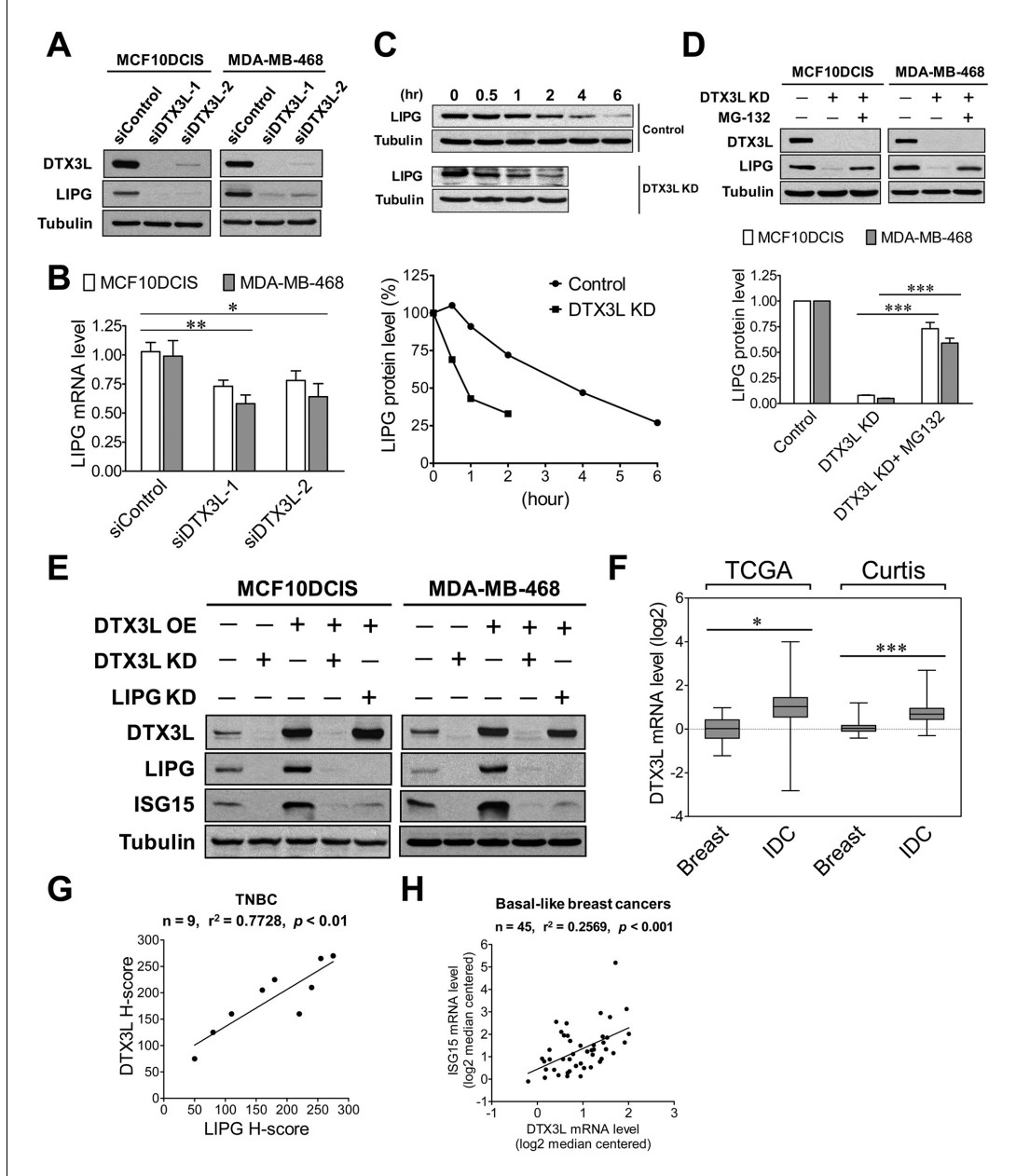

**Figure 7.** DTX3L is the upstream activator of the LIPG-ISG15 signaling axis in basal-like TNBC cells. (A) DTX3L inactivation leads to a dramatic reduction in LIPG protein expression in MCF10DCIS and MDA-MB-468. Western blot analysis of DTX3L, LIPG and tubulin was performed on MCF10DCIS and MDA-MB-468 cells with or without DTX3L knockdown by two different siRNAs. (B) DTX3L knockdown results in a modest decrease in LIPG mRNA levels in MCF10DCIS and MDA-MB-468 cells. qRT-PCR analysis of LIPG mRNA expression was performed on siRNA-transfected cells as described in (A). (C) DTX3L deficiency shortens the half-life of LIPG protein in MDA-MB-468 cells. After treatment with the protein synthesis inhibitor cycloheximide (200 μg/ml), MDA-MB-468 cells with or without DTX3L knockdown were harvested at different time points as indicated for western blot analysis of LIPG and tubulin. Kinetic changes in LIPG protein levels during the time course are shown in the quantitative plot on the bottom panel. (D) DTX3L depletion facilitates proteasome-mediated LIPG protein degradation. Western blot analysis of DTX3L, LIPG and tubulin was performed on control and DTX3L-knockdown MDA-MB-468 cells with or without MG-132 treatment (10 μM) for 10 hr. The quantitative data are shown on the bottom panel. (E) The DTX3L-LIPG signaling axis is required for ISG15 expression in MCF10DCIS and MDA-MB-468 cells. Western blot analysis of DTX3L, LIPG, ISG15 and tubulin was performed on MCF10DCIS and MDA-MB-468 cells with or without DTX3L knockdown, and on their respective DTX3L-overexpressing cells with or without knockdown of either DTX3L or LIPG. (F) DTX3L is aberrantly overexpressed in breast cancer. DTX3L mRNA expression in normal breast and IDC was analyzed *in silico* using the TCGA and Curtis datasets. TCGA: breast (n = 61), IDC (n = 389); Curtis: breast (n = 144), IDC (n = 1556). The 25th and 75th percentiles are indicated as a vertical box and the minimal and maximal data values are indicated as outliers. (G) DTX3L protein expression positively correlates with LIPG expression in TNBC. IHC analysis of DTX3L and LIPG was performed on nine

*Figure 7 continued*

cases of TNBC. The linear regression correlation was plotted based on H-scores of DTX3L and LIPG IHC-stained TNBC cases. (**H**) DTX3L mRNA expression is positively associated with ISG15 expression in human basal-like breast cancers. The linear regression correlation was plotted based on *in silico* analysis of DTX3L and ISG15 expression in 45 cases of BLBCs derived from the Gluck dataset. Error, SD (n = 3); *p<0.05; **p<0.01; ***p<0.001.

DOI: https://doi.org/10.7554/eLife.31334.030

The following source data and figure supplements are available for figure 7:

**Source data 1.** Source data for *Figure 7*.

DOI: https://doi.org/10.7554/eLife.31334.034

**Source data 2.** DTX3L expression in breast cancer.

DOI: https://doi.org/10.7554/eLife.31334.035

**Source data 3.** DTX3L expression is positively associated with LIPG and ISG15.

DOI: https://doi.org/10.7554/eLife.31334.036

**Source data 4.** DTX3L expression in different molecular subtypes of breast cancer.

DOI: https://doi.org/10.7554/eLife.31334.037

**Figure supplement 1.** Western blot analysis of DTX3L and tubulin protein expression in MCF10DCIS and MDA-MB-468 cells with or without knockdown of either LIPG or ISG15.

DOI: https://doi.org/10.7554/eLife.31334.031

**Figure supplement 2.** Expression of DTX3L mRNA in basal-like (n = 45), HER2+ (n = 21), luminal-A (n = 46) and luminal-B (n = 25) subtypes of breast cancer.

DOI: https://doi.org/10.7554/eLife.31334.032

**Figure supplement 3.** LIPG protein expression is positively associated with DTX3L protein expression in TNBCs.

DOI: https://doi.org/10.7554/eLife.31334.033

with the proteasome inhibitor MG-132. MG-132 treatment restored approximately 75% and 60% of control LIPG protein levels in DTX3L-knockdown MCF10DCIS and MDA-MB-468 cells, respectively (*Figure 7D*). These findings indicate that DTX3L is required for protecting LIPG from protein degradation in basal-like TNBC cells.

To determine whether DTX3L functions as an activator upstream of the LIPG-ISG15 signaling pathway, we performed gain- and loss-of-function studies of DTX3L in combination with LIPG knockdown. As shown in *Figure 7E*, knockdown of endogenous DTX3L dramatically downregulated the protein expression of LIPG and ISG15 in both MCF10DCIS and MDA-MB-468, whereas exogenous DTX3L overexpression enhanced expression of LIPG and ISG15 in both cell lines. Furthermore, similar to the DTX3L-knockdown effect on ISG15, LIPG knockdown completely abolished the induction of ISG15 expression in DTX3L-overexpressing MCF10DCIS and MDA-MB-468 cells without affecting the protein levels of overexpressed DTX3L (*Figure 7E*), demonstrating that LIPG mediates the DTX3L-dependent effect to increase ISG15 expression. Knockdown of either LIPG or ISG15 had no impact on endogenous DTX3L protein levels in MCF10DCIS and MDA-MB-468 cells (*Figure 7—figure supplement 1*). These findings, taken together, indicate that DTX3L functions as an upstream regulator activating the LIPG-ISG15 signaling axis.

The expression and functional role of DTX3L in breast cancer have not yet been explored. To reveal the expression of DTX3L in breast cancer tissues, we conducted *in silico* analysis of DTX3L mRNA expression in breast cancer using the Oncomine database. The analyses of two breast cancer datasets, TCGA and Curtis (*Curtis et al., 2012*), indicated that DTX3L was overexpressed in IDC compared to normal breast (*Figure 7F*). Differential expression of DTX3L was not significant among different molecular subtypes of breast cancer (*Figure 7—figure supplement 2*), suggesting that DTX3L may play a role in other breast cancer subtypes in addition to its role in basal-like TNBC. To determine whether DTX3L expression is positively associated with LIPG expression in TNBC, IHC studies were performed on nine cases of TNBC. Some representative IHC staining pictures are shown in *Figure 7—figure supplement 3*. After H-scoring of IHC staining data, we found that DTX3L protein staining was positively correlated with LIPG protein staining (p<0.01) (*Figure 7G*). DTX3L mRNA levels were also positively associated with ISG15 mRNA levels (p<0.001) according to *in silico* analysis of BLBC cases (n = 45) (*Figure 7H*). These findings suggest the existence of the DTX3L-LIPG-ISG15 signaling pathway in human basal-like TNBC.

## The oncogenic DTX3L-LIPG-ISG15 signaling axis is implicated in promoting basal-like TNBC development

To reveal the functional role of DTX3L in basal-like TNBC, we first conducted DTX3L knockdown studies on MCF10DCIS cells. As shown in *Figure 8A*, DTX3L knockdown by either one of two siRNAs profoundly impaired cell growth as well as CSC sphere formation and abrogated migratory and invasive abilities of MCF10DCIS cells. Consistently, DTX3L inactivation resulted in an approximately two-fold increase in the EpCAM+ cell population, decreased expression of stem-cell and basal/EMT programming genes and the reduced EMT (E-cadherin upregulation and vimentin downregulation) in MCF10DCIS cells (*Figure 8B, C, D and E*). DTX3L knockdown also led to downregulation of vimentin protein expression in MDA-MB-468 cells (*Figure 8F*). These data indicate that DTX3L is required for invasiveness, stemness and basal/EMT features of TNBC.

We next sought to investigate whether LIPG and ISG15 mediate the functional effects of DTX3L in basal-like TNBC cells. To address this question, we examined the impact of knocking down either LIPG or ISG15 in DTX3L-overexpressing MDA-MB-468 cells. Exogenous DTX3L overexpression promoted invasion, CSC sphere formation and expression of stem-cell and EMT genes, whereas DTX3L knockdown completely abolished these phenotypes (*Figure 8G, H and I*), demonstrating that these observed events were DTX3L-specific. LIPG knockdown also effectively abolished all of these DTX3L-overexpression phenotypes (*Figure 8G, H and I*), indicating that LIPG mediates the enhancing effect of DTX3L on invasiveness, stemness and basal/EMT features of MDA-MB-468 cells. Moreover, ISG15 knockdown suppressed DTX3L-overexpression-promoted invasion more significantly than its inhibitory effect on CSC sphere formation (*Figure 8G and H*). These results corroborated the gene expression data which showed that ISG15 knockdown inhibited DTX3L-mediated expression of EMT genes more potently than it suppressed the elevated expression of stem-cell genes (*Figure 8I*). Xenograft studies further demonstrated that similar to DTX3L knockdown, knockdown of either LIPG or ISG15 completely abolished the DTX3L overexpression-enhanced growth of MDA-MB-468 xenograft tumors and also significantly inhibited tumor development (*Figure 8J*). These *in vitro* and *in vivo* studies suggest that the DTX3L-LIPG-ISG15 signaling axis is pivotal for the development of basal-like TNBC.

## Discussion

In this study, we have identified that LIPG is aberrantly overexpressed in a subset of basal-like TNBCs. We have further demonstrated that LIPG is crucial for malignant characteristics of LIPG-expressing TNBC cells, including invasiveness, stemness, basal/EMT features, *in vivo* tumorigenicity and *in vivo* metastasis. Moreover, LIPG is required for maintaining basal/EMT and tumorigenic phenotypes of the basal-like TNBC line MCF10DCIS, which forms basal-like ductal carcinoma *in situ* (DCIS) *in vivo* (*Miller et al., 2000*; *Hu et al., 2008*). Given that basal-like DCIS has been considered to be the precursor of invasive TNBC (*Bryan et al., 2006*; *Thike et al., 2013*), our studies suggest that LIPG may play an oncogenic role in the progression from basal-like DCIS to invasive TNBC.

Although it has been proposed that the phospholipase activity of LIPG and its function in importing lipid precursor species into cells are involved in promoting breast cancer cell proliferation (*Slebe et al., 2016*), the exact role of LIPG in breast cancer remains largely unknown. By studying the catalytically inactive mutant of LIPG, we for the first time have revealed that LIPG possess both lipase-dependent and lipase-independent functions in breast cancer cells. Our studies show that the lipase function of LIPG is implicated in supporting cell growth and promoting cell proliferation rate. In contrast, the lipase-independent function is required for LIPG to enhance invasiveness, stemness and basal/EMT features of breast cancer cells. This natural design of dual functional roles for LIPG is necessary as cell proliferation is not always coupled with the EMT and active in cancer stem cells. For example, TGFβ signaling has been known to activate the EMT and induce growth suppression in a parallel manner (*Tsuji et al., 2008*), and cancer stem cells can undergo dormancy during metastasis and chemoresistance (*Elshamy and Duhé, 2013*; *Giancotti, 2013*; *Pinto et al., 2013*; *Sosa et al., 2014*). Therefore, these two identified functional aspects of LIPG allow cancer cells to have more plasticity in regulating cell proliferation, stemness and EMT. It would be important in the future to reveal how cancer cells separately regulate enzymatic and non-enzymatic functions of LIPG during their tumorigenesis, metastasis and chemoresistance.

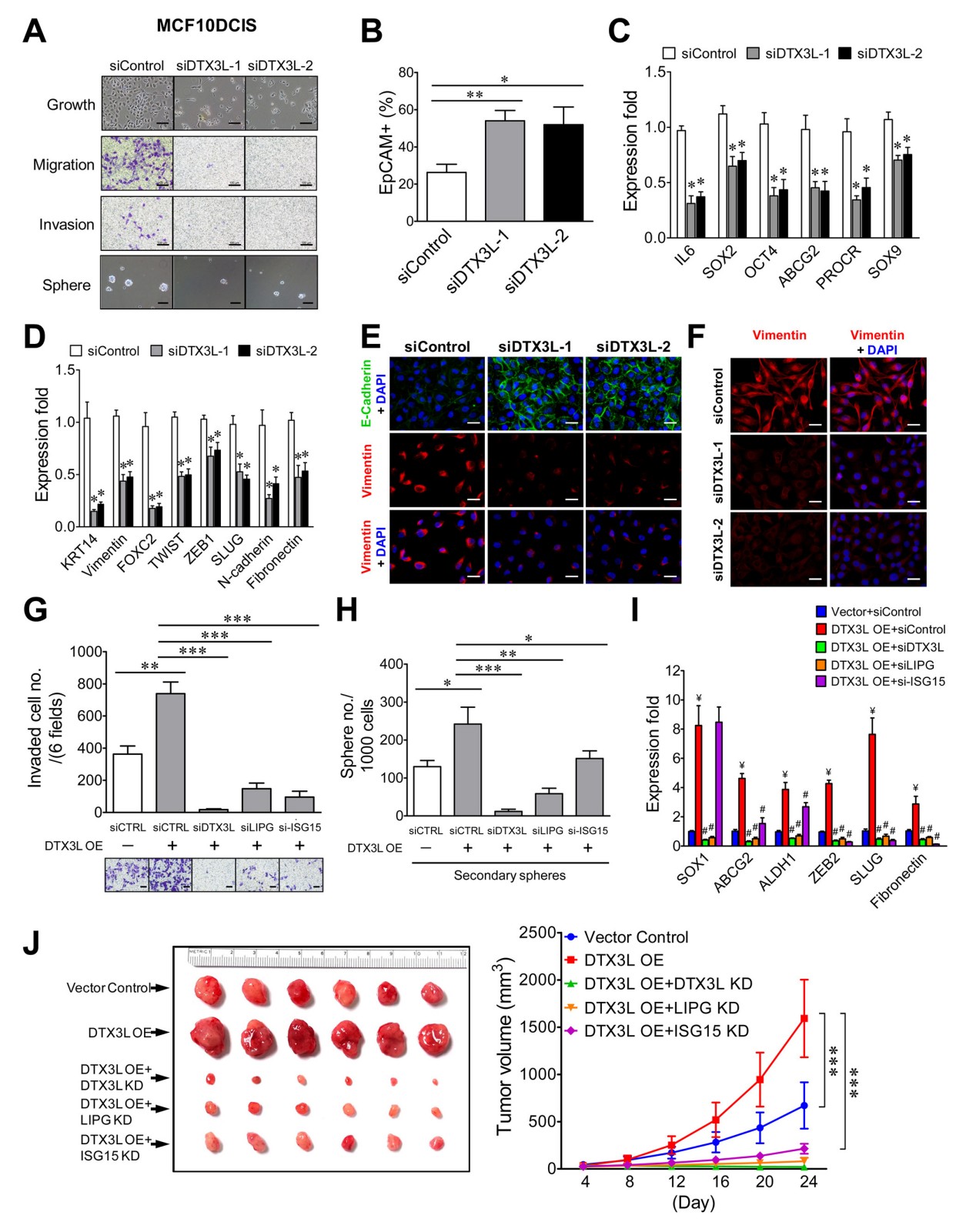

**Figure 8.** The DTX3L-LIPG-ISG15 signaling axis is essential for malignancies of LIPG-expressing TNBC cells *in vitro* and *in vivo*. (**A**) DTX3L knockdown inhibits growth, migration, invasion and CSC sphere formation of MCF10DCIS cells. The scale bars indicate 100 μm. (**B**) DTX3L knockdown increases EpCAM+ cells in MCF10DCIS. (**C**) DTX3L inactivation downregulates the expression of multiple stem cell genes in MCF10DCIS cells. (**D**) DTX3L depletion induces the decreased expression of multiple basal/EMT genes in MCF10DCIS cells. (**E**) DTX3L knockdown leads to E-cadherin upregulation

*Figure 8 continued on next page*

*Figure 8 continued*

and vimentin downregulation in MCF10DCIS cells. The scale bar indicates 25 μm. (F) DTX3L knockdown results in vimentin downregulation in MDA-MB-468 cells. The scale bar indicates 25 μm. (G) The LIPG-ISG15 axis mediates DTX3L overexpression-enhanced invasion of MDA-MB-468 cells. Transwell-based invasion analysis was performed on DTX3L-overexpressing MDA-MB-468 cells with or without siRNA-mediated knockdown of a gene component in the DTX3L-LIPG-ISG15 axis and parental control cells. (H) The LIPG-ISG15 axis mediates DTX3L overexpression-promoted secondary CSC sphere formation of MDA-MB-468 cells. Secondary CSC sphere formation analysis was performed on cells as described in (G). (I) The LIPG-ISG15 axis mediates DTX3L overexpression-induced increases in expression of stem cell and EMT genes in MDA-MB-468 cells. qRT-PCR analysis of three stem cell and three EMT genes was performed on cells as describe in (G). ¥ p<0.01 versus the control dataset (Vector + siControl); # p<0.05 versus the dataset of MDA-MB-468-DTX3L OE cells transfected with siControl (DTX3L OE +siControl). (J) The DTX3L-LIPG-ISG15 axis is required for *in vivo* tumor formation of MDA-MB-468 cells in nude mice. 1 × 10$^6$ of DTX3L-overexpressing MDA-MB-468 cells with or without shRNA-mediated knockdown of a gene component in the DTX3L-LIPG-ISG15 axis or parental control cells were transplanted into the mammary fat pads of nude mice for xenograft tumor formation. The picture of isolated tumors and plotted tumor growth curves are shown on the left and right panels, respectively. Error, SD (n = 3); *p<0.05; **p<0.01; ***p<0.001.

DOI: https://doi.org/10.7554/eLife.31334.038

The following source data is available for figure 8:

**Source data 1.** Source data for *Figure 8.*
DOI: https://doi.org/10.7554/eLife.31334.039

Mechanistically, we have found that LIPG executes its oncogenic function in basal-like TNBC cells through its involvement in the DTX3L-ISG15 pathway, known for mediating IFN signaling. This finding for the first time links the function of a lipase to a pathway relevant in immune response and cancer development. We have characterized the functional relationship between these three genes and defined the DTX3L-LIPG-ISG15 signaling axis in basal-like TNBC cells. Moreover, our novel finding indicates that DTX3L is required to sustain LIPG expression in basal-like TNBC cells mainly through its protection of LIPG from proteasome-mediated protein degradation. We do not yet understand how DTX3L executes this protective function to maintain LIPG protein levels in TNBC cells. However, it has been shown that DTX3L is involved in the immunoproteasome pathway to degrade viral 3C proteases (*Zhang et al., 2015b*). Therefore, there is a possibility that DTX3L protects LIPG protein levels via degrading some cellular proteases involved in LIPG protein degradation. Future study is necessary to address this possibility. From the study of a catalytically inactive LIPG mutant, we have revealed that the LIPG induces ISG15 expression in a non-enzymatic manner. This finding suggests that the non-enzymatic function of LIPG is implicated in the DTX3L-LIPG-ISG15 signaling pathway. Several cellular enzymes (e.g. histone H3 K36 demethylase KDM4A, caspase-8, and lysyl oxidase) have been reported to possess both enzymatic and non-enzymatic functions (*Trackman, 2016*; *Colmenares et al., 2017*; *Henry and Martin, 2017*). Most of these identified non-enzymatic functions are mediated by protein-protein interactions. These discoveries raise a possibility that the non-enzymatic function of LIPG is mediated by its interacting proteins. Due to the critical role of the DTX3L-LIPG-ISG15 axis in tumorigenicity of basal-like TNBC cells *in vitro* and *in vivo* and the essential role of LIPG in TNBC metastasis *in vivo*, future studies are needed to address several important questions raised by our findings. What remains unclear is how the non-enzymatic role of LIPG can be engaged in IFN signaling and regulate expression of ISGs, how the DTX3L-LIPG-ISG15 axis regulates the malignancies of basal-like TNBC and what other factors are involved, and what pathological mechanisms cause LIPG mRNA overexpression in basal-like TNBC.

To our surprise, LIPG overexpression in MCF7 cells increased expression of many ISGs related to type I and II IFN responses. This finding suggests that LIPG is involved in IFN signaling in breast cancer cells by enhancing expression of ISGs. It will be important in the future to reveal whether LIPG participates in IFN responses in immune cells and other cell types. It has been shown that LIPG expression can be induced by immune cytokines (e.g. IL-6), agonists (e.g. polyinosinic:polycytidylic acid (poly I:C) and lipopolysaccharide (LPS)) and transcription factors (e.g. NFκB) (*Kempe et al., 2005*; *Wang et al., 2007*; *Yasuda et al., 2007*; *Badellino et al., 2008*; *Robert et al., 2013*). However, the biological roles of LIPG in these immune responses remain unclear. Our studies have shown that DTX3L, a ubiquitin E3 ligase essential for IFN-stimulated expression of ISGs, is required for protecting LIPG protein from proteasome-mediated degradation in breast cancer cells. Given that IFNs can induce DTX3L expression (*Juszczynski et al., 2006*; *Zhang et al., 2015b*), IFNs may enhance LIPG expression in cancer cells through their stimulating effects on DTX3L expression. This potential

connection is relevant as IFN-β has been shown to be necessary and sufficient for increased tumorigenicity in a transformed mouse embryonic fibroblast line (*Forys et al., 2014*). Because this tumorigenic cell model has defects in both p53 and ARF, it is akin to TNBC, which frequently manifests co-inactivation of both tumor suppressor genes (*Forys et al., 2014*). Moreover, knockdown of STAT1, an important mediator of IFN signaling, impaired growth of TNBC cells (*Forys et al., 2014*). These findings suggest that IFN signaling may play a critical role in TNBC development. Although activation of the IFN signature is generally thought to be tumor-suppressive (*Chan et al., 2012*; *Snell et al., 2017*), our and other studies suggest that IFN-related signaling pathways function as double-edged swords in cancer (*Forys et al., 2014*; *Snell et al., 2017*). Our finding of DTX3L-dependent regulation of LIPG expression may have important implications to tumor-microenvironmental interactions occurring in TNBCs through secreted immune cytokines like IFNs.

Our finding that LIPG is preferentially overexpressed in basal-like TNBCs is inconsistent with the previous finding by Slebe et al., which shows that LIPG is generally overexpressed in all subtypes of breast cancer (*Slebe et al., 2016*). This discrepancy is likely due to variations in the use of the LIPG antibody, cell line models, breast cancer datasets, etc. The most obvious difference is that their antibody only detected cleaved 40 kDa LIPG (*Slebe et al., 2016*). Contrary to their result, our antibody detected both 68 kDa and 40 kDa LIPG proteins, consistent with other reported findings (*Edmondson et al., 2009*). Our studies showed that 68 kDa glycosylated LIPG is the predominant form in breast cancer cells. Our findings from studying LIPG protein expression in breast cancer tissues and cell lines are consistent with the results from *in silico* analyses of public breast cancer datasets from the Oncomine database.

In conclusion, our studies highlight the importance of the DTX3L-LIPG-ISG15 signaling pathway in the tumorigenesis and metastasis of basal-like TNBC and the potential to target this signaling axis for therapy of basal-like TNBC expressing LIPG. Moreover, these novel findings suggest a new view that a lipoprotein lipase can influence protein functionality in a non-enzymatic manner via its impact on expression of ISG15, a ubiquitin-like protein involved in regulating protein function and stability by ISGylation. We predict that the regulation of DTX3L-ISG15 signaling may be different between LIPG-proficient and LIPG-deficient cancer cells. Understanding of signaling alterations between these two scenarios would propel the identification of effective therapeutic targets for TNBC therapy.

## Materials and methods

**Key resources table**

| Reagent type (species) or resource | Designation | Source or reference | Identifiers | Additional information |
|---|---|---|---|---|
| Gene (human) | LIPG | National Center for Biotechnology Information (https://www.ncbi.nlm.nih.gov/gene/9388) | Gene ID: 9388; Accession number: NM_006033; UniPro ID: Q9Y5 × 9 | |
| Gene (human) | DTX3L | National Center for Biotechnology Information (https://www.ncbi.nlm.nih.gov/gene/151636) | Gene ID: 151636; Accession number: NM_138287; UniPro ID: Q8TDB6 | |
| Gene (human) | ISG15 | National Center for Biotechnology Information (https://www.ncbi.nlm.nih.gov/gene/9636) | Gene ID: 9636; Accession number: NM_005101 | |
| Strain, strain background (mouse) | Athymic nude (nu/nu) mice | The Jackson Laboratory (https://www.jax.org/strain/002019) | Stock No: 002019 | |
| Cell line (human) | MDA-MB-231 | ATCC (https://www.atcc.org/Products/All/HTB-26.aspx) | Catalog number: ATCC HTB-26; RRID:CVCL_0062 | |
| Cell line (human) | MDA-MB-468 | ATCC (https://www.atcc.org/Products/All/HTB-132.aspx) | Catalog number: ATCC HTB-132; RRID:CVCL_0419 | |
| Cell line (human) | Hs578T | ATCC (https://www.atcc.org/Products/All/HTB-126.aspx) | Catalog number: ATCC HTB-126; RRID:CVCL_0332 | |
| Cell line (human) | MCF10DCIS.COM | Asterand Bioscience (https://www.asterandbio.com) | RRID:CVCL_5552 | |

*Continued on next page*

*Continued*

| Reagent type (species) or resource | Designation | Source or reference | Identifiers | Additional information |
|---|---|---|---|---|
| Cell line (human) | MCF7 | ATCC (https://www.atcc.org/Products/All/HTB-22.aspx) | Catalog number: ATCC HTB-22; RRID:CVCL_0031 | |
| Cell line (human) | T47D | ATCC (https://www.atcc.org/Products/All/HTB-133.aspx) | Catalog number: ATCC HTB-133; RRID:CVCL_0553 | |
| Cell line (human) | HEK293T | ATCC (https://www.atcc.org/Products/All/CRL-3216.aspx) | Catalog number: ATCC CRL-3216; RRID:CVCL_0063 | |
| Antibody | Mouse anti-human LIPG monoclonal antibody | Abcam | Catalog number: ab56493; RRID:AB_943999 | Applications: WB, IHC |
| Antibody | Rabbit anti-human DTX3L polyclonal antibody | Bethyl Laboratories, Inc | Catalog number: A300-834A; RRID:AB_2277461 | Applications: WB |
| Antibody | Rabbit anti-human DTX3L polyclonal antibody | Bethyl Laboratories, Inc | Catalog number: A300-833A; RRID:AB_597865 | Applications: IHC, IP |
| Antibody | Rabbit anti-human vimentin polyclonal antibody | Proteintech | Catalog number: 10366–1-AP; RRID:AB_2273020 | Applications: IHC |
| Antibody | Rabbit anti-human ISG15 polyclonal antibody | Cell Signaling Technology | Catalog number: #2743; RRID:AB_2126201 | Applications: WB |
| Antibody | Rabbit anti-human ISG15 polyclonal antibody | Cell Signaling Technology | Catalog number: #2758; RRID:AB_2126200 | Applications: WB |
| Antibody | Mouse anti-human $\alpha$-tubulin monoclonal antibody | Thermo Fisher Scientific | Catalog number: 13–8000; RRID:AB_2533035 | Applications: WB |
| Antibody | Rat anti-human CD24 monoclonal antibody, phycoerythrin (PE)-conjugated | BioLegend | Catalog number:311106; RRID:AB_314854 | Applications: Flow Cytometry |
| Antibody | Rat anti-mouse/human CD44 monoclonal antibody, allophycocyanin (APC)-conjugated | BioLegend | Catalog number: 103012; RRID:AB_312963 | Applications: Flow Cytometry |
| Antibody | Mouse anti-human CD326 (EpCAM) monoclonal antibody, allophycocyanin (APC)-conjugated | BioLegend | Catalog number: 324208; RRID:AB_756081 | Applications: Flow Cytometry |
| Antibody | Mouse anti-human E-cadherin monoclonal antibody | Thermo Fisher Scientific | Catalog number: MA1-34228; RRID:AB_1955688 | Applications: Immunofluorescence |
| Antibody | Mouse anti-human vimentin monoclonal antibody | Thermo Fisher Scientific | Catalog number: MA1-33444; RRID:AB_2537032 | Applications: Immunofluorescence |
| Recombinant DNA reagent | pSMPUW-Puro Lentivirus Expression Vector | Cell Biolabs, Inc | Catalog number: VPK-212 | |
| Chemical compound, drug | Cycloheximide | Sigma-Aldrich | Catalog number: C1988 | |
| Chemical compound, drug | MG-132 | Sigma-Aldrich | Catalog number: 474787 | |
| Software, algorithm | The GraphPad Prism software | GraphPad Software, Inc | RRID:SCR_002798 | |

## Cell culture

We obtained breast cancer cell lines, including MDA-MB-231 (RRID:CVCL_0062), MDA-MB-468 (RRID:CVCL_0419), Hs578T (RRID:CVCL_0332), T47D (RRID:CVCL_0553) and MCF7 (RRID:CVCL_0031), from ATCC (American Type Culture Collection, Manassas, VA). The human DCIS cell line MCF10DCIS.COM (MCF10DCIS) (RRID:CVCL_5552) was purchased from Asterand USA (Detroit, MI). These cell lines were cultured according to manufacturer's instructions. The identity of cell lines has been authenticated by ATCC and Asterand before we obtained them. We routinely perform mycoplasma contamination testing using the mycoplasma detection kit (ATCC) to ensure that cell lines used in studies are mycoplasma-free.

### *In silico* analysis of gene expression

The Oncomine Cancer Microarray Database (RRID:SCR_007834) (http://www.oncomine.org) (*Rhodes et al., 2007*) was used to perform *in silico* expression analysis of *LIPG*, *DTX3L* and *ISG15* genes in normal and cancerous breast tissues.

## Normal and cancerous breast tissues

The adjacent normal and cancerous breast tissue samples from breast cancer patients (38 to 80 years old with a mean age of 51 ± 11 years) who underwent surgery during the year 2016 were provided by the Chonnam National University Hwasun Hospital National Biobank of Korea, a member of the National Biobank of Korea, which is supported by the Ministry of Health, Welfare and Family Affairs. Subtype classification of invasive ductal carcinomas used in the study was based on their IHC staining results of ER, PR and HER2. All tissue samples were obtained with informed consent from patients under protocols approved by the institutional review board of the Chonnam National University Hwasun Hospital. These obtained tissue specimens were utilized to generate tissue section slides and tissue microarrays for research purposes. The use of human tissue specimens in this study has been approved by the institutional review board of the Chonnam National University Hwasun Hospital (Reference number: CNUHH-2016–153).

## RNA isolation and quantitative RT-PCR (qRT-PCR) analysis

Total RNA of cultured cells was isolated using the Ambion TRIzol reagent (Thermo Fisher Scientific, Halethorpe, MD) and total RNA of paraffin-embedded breast tissues was isolated using the AllPrep DNA/RNA FFPE Kit (Qiagen, Hilden, Germany) according to manufactures' instructions. qRT-PCR analysis of mRNA expression was performed as described previously with normalization to GAPDH (*Lo et al., 2016*). The gene primers used in studies are listed as following:

*LIPG*: forward, 5'-TGGACTCAACGATGTCTTGG-3'
reverse, 5'-AACTCGGCTTGTCCTGATTC-3'
*ISG15*: forward, 5'-GCCATGGGCTGGGACCT-3'
reverse, 5'-TGATCTGCGCCTTCAGCTCT-3'
*KRT14*: forward, 5'-CAGTCCCAGCTCAGCATGAA-3'
reverse, 5'-CCACGCTGCCAATCATCTC-3'
*CDH1*/E-cadherin: forward, 5'- GCCCTTGGAGCCGCAG-3'
reverse, 5'-TCAAAATTCACTCTGCCCAGGA-3'
*CDH2*/N-cadherin: forward, 5'-TATCCTTGTGCTGATGTTTGTGGT-3'
reverse, 5'-GCTCAAGTCATAGTCCTGGTCT-3'
*FOXC2*: forward, 5'-CAGCAGCAAACTTTCCCCAAC-3'
reverse, 5'-CAGTATTTCGTGCAGTCGTAGGAGTAG-3'
*SNAI1/SNAIL*: forward, 5'-GCCTAGCGAGTGGTTCTTCTG-3'
reverse, 5'-CTGCTGGAAGGTAAACTCTGG-3'
*SNAI2/SLUG*: forward, 5'-ACTGGACACACATACAGTGATT-3'
reverse, 5'- ACTCACTCGCCCCAAAGATG-3'
*TWIST1*: forward, 5'-GCCGGAGACCTAGATGTCATT-3'
reverse, 5'-CCCACGCCCTGTTTCTTTGA-3'
*ZEB1*: forward, 5'-GAGACATAAATATGAACACACAGGTAAAAGAC-3'
reverse, 5'-TTGAGAATAAGACCCAGAGTGTGAGAAG-3'
*ZEB2*: forward, 5'-CCACCAGTCCAGACCAGTATTCCT-3'
reverse, 5'-CATCAAGCAATTCTCCCTGAAATC-3'
*FN1*/Fibronectin 1: forward, 5'-AGATACCATCAGAGAACAAACACTAA-3'
reverse, 5'-GATGGATCTTGGCAGAGAGACA-3'
*VIM*/vimentin: forward, 5'-CGGGAGAAATTGCAGGAGGA-3'
reverse, 5'-AAGGTCAAGACGTGCCAGAG-3'
*IL6*: forward, 5'-CAATCTGGATTCAATGAGGAGACTT-3'
reverse, 5'-TACTCTCAAATCTGTTCTGGAGGT-3'
*SOX1*: forward, 5'-AGTGTCGCTCCAATTCAAATTAGTG-3'
reverse, 5'-GATTTGGGAAGTGAATGAAGTCGT-3'
*SOX2*: forward, 5'-GCTTTTGTTCGATCCCAACTTTC-3'

reverse, 5'-ATGGATTCTCGGCAGACTGATTC-3'
*SOX9*: forward, 5'-TCAACGGCTCCAGCAAGAACAAG-3'
reverse, 5'-ACTTGTAATCCGGGTGGTCCTTCT-3'
*POU5F1/OCT4*: forward, 5'-CGAAAGAGAAAGCGAACCAGTATC-3'
reverse, 5'-AGAACCACACTCGGACCACATC-3'
*ABCG2*: forward, 5'-TTTTCAGGTCTGTTGGTCAATCTCA-3'
reverse, 5'-CATTATGCTGCAAAGCCGTAAATC-3'
*PROCR*: forward, 5'-CAGACACCAACACCACGATCA-3'
reverse, 5'-CAGCGGATGGTCAGAGGAAAG-3'
*ALDH1*: forward, 5'-ATGTCTGGAAATGGAAGAGAACTGG-3'
reverse, 5'-GTGACTGTAAGGAGATGCTTAGCTATTGAA-3'
*PROM1/CD133*: forward, 5'-CACAGGGAATGGATTGTTGGA-3'
reverse, 5'-CACGATGCCACTTTCTCACTGATAG-3'

## Immunohistochemistry assay

The immunohistochemistry (IHC) analysis was performed using the avidin biotin peroxidase complex (ABC) method as previously described (*Lo et al., 2017*). The antibodies used in IHC experiments include anti-LIPG (Abcam, Cambridge, United Kingdom; catalog no. ab56493) (RRID:AB_943999), anti-DTX3L (Bethyl Laboratories, Inc, Montgomery, TX; catalog no. A300-833A) (RRID: AB_597865) and anti-vimentin (Proteintech, Rosemont, IL; catalog no. 10366–1-AP) (RRID:AB_2273020). Scoring of LIPG and DTX3L protein expression in human breast cancer tissue samples was scored independently by two individuals based on an H-score, which is derived by multiplying the staining intensity (0–3) with the percentage of tumor epithelial cells with positive IHC staining.

## Western blot analysis

Protein expression of LIPG, ISG15 and DTX3L was examined by western blotting using a mouse antibody against LIPG (Abcam, ab56493) (RRID:AB_943999) and rabbit antibodies against ISG15 (Cell Signaling Technology, #2758 and #2743) (RRID:AB_2126200 and RRID:AB_2126201) and DTX3L (Bethyl Laboratories, Inc, A300-834A) (RRID: AB_2277461). Protein expression was detected by chemiluminescence (ECL, Pierce). Expression of α-tubulin (Thermo Fisher Scientific) (RRID:AB_2533035) was used as a protein loading control. Western blot data were quantified by densitometric analysis of autoradiograms, using a computerized densitometer (Typhoon System; Molecular Dynamics, Inc., Sunnyvale, CA). The quantitative protein level data were normalized by the α-tubulin protein levels.

## Establishment of stable MCF7 and MDA-MB-468 lines with gene overexpression and/or knockdown

The full-length of human LIPG cDNA fragments were amplified from the cDNA pool prepared from MCF10DCIS RNA. To generate lenti-CMV-LIPG, the LIPG cDNA was cloned into the lentiviral destination vector pSMPUW-CMV (*Zhu et al., 2015*) using the Gateway LR cloning kit (Thermo Fisher Scientific) according to manufacturer's instructions. To generate a catalytically inactive mutant of LIPG, we introduced two nucleotide mutations into the *LIPG* gene cDNA within the pSMPUW-CMV vector for converting serine at the 149 aa of matured LIPG protein (lacking the N-terminal 20-aa signal peptide) to alanine (S149A) using a QuikChange Lightning Site-Directed Mutagenesis Kit (Agilent, Santa Clara, CA) according to manufacturer's instructions.

To produce lentiviruses, 1.2 μg of lenti-CMV-LIPG (or lenti-CMV-LIPG$^{S149A}$), 1.2 μg of pCD/NL-BH*DDD (Addgene, Cambridge, MA; plasmid 17531) and 0.2 μg of pVSVG (Cell Biolabs, San Diego, CA) were co-transfected into HEK293T cells (RRID:CVCL_0063) using Lipofectamine 2000 (Thermo Fisher Scientific) in a six-well plate. The Lipofectamine 2000/DNA complex-containing medium was replaced with the fresh medium (DMEM supplemented with 10% FBS and 100 U/ml penicillin-streptomycin) 16 hr post-transfection. Cell medium containing the viral particles was collected twice at 24 and 48 hr after the medium change, passed through a 0.45 μm filter, and either used fresh or stored at 4°C for up to 1 week for cell infection.

To establish stable LIPG-overexpressing MCF7 and MDA-MB-468 cells, both cell lines were infected by exposure to the lentiviral supernatant (~4 MOI) and polybrene (0.8 μg/mL) (Millipore,

Billerica, MA) daily for 2 days. 24 hr after the second lentiviral infection, infected cells were selected in the culture medium containing puromycin (0.8 µg/ml) (Sigma-Aldrich, St. Louis, MO).

To construct the human DTX3L expression plasmid, the full-length *DTX3L* gene cDNA fragments were amplified from the cDNA pool prepared from MCF10DCIS RNA. The *DTX3L* cDNA was cloned into the BamHI and XbaI sites of the expression vector pcDNA3.1. To establish the stable DTX3L-overexpressing MDA-MB-468 cell line, cells were transfected with pcDNA3.1-DTX3L using Lipofect-amine 2000 and transfected cells were selected in the culture medium containing 1 mg/ml geneticin (G418) (Sigma-Aldrich, St. Louis, MO).

The shRNA sequences were cloned into pLV-hU6-Ef1a-puro vector (Biosettia, San Diego, CA) according to manufacturer's instructions. The scramble and four shRNA sequences used in the study are: scramble shRNA, 5'-TAACTCGCTCGAAGGAATC-3'; shLIPG-1, 5'-GCCTTTCAGAGTTTACCAT-3'; shLIPG-2, 5'-GCCGCAAGAACCGTTGTAA-3'; shISG15, 5'-GCAACGAATTCCAGGTGTC-3'; shDTX3L, 5'-GGCGAAGCAGTATGTTCTA-3'. The preparation of lentiviral shRNA particles and cell infection were processed as described above. To establish stable gene knockdown cell lines, infected MDA-MB-468 cells with or without DTX3L overexpression were selected in the culture medium containing puromycin (0.8 µg/ml).

## siRNA transfection

The siRNA transfection was performed with 40 nM of siRNA using Lipofectamine RNAiMAX (Thermo Fisher Scientific) according to manufacturer's instructions. siRNAs were obtained from Sigma-Aldrich. The siRNA sequences used in the study are: siControl, 5'-UAACUCGCUCGAAGGAAUC-3' (*Lo et al., 2016*); siLIPG-1, 5'-GCCUUUCAGAGUUUACCAU-3' (*Slebe et al., 2016*); siLIPG-2, 5'-GCCGCAAGAACCGUUGUAA-3'; si-ISG15, 5'-GCAACGAAUUCCAGGUGUC-3' (*Zhang et al., 2015a*); siDTX3L-1, 5'-GGCGAAGCAGUAUGUUCUA-3' (*Juszczynski et al., 2006*); siDTX3L-2, 5'-GAGGUGGGUCCGAAAUAAU-3' (*Bachmann et al., 2014*).

## Migration and invasion assays

Transwell-based migration and invasion assays were implemented as previously described (*Lo et al., 2016*).

## Stem-cell sphere formation assay

We performed primary stem-cell sphere formation assays as previously described (*Lo et al., 2016*). For secondary sphere formation assays, formed primary spheres were collected and dissociated with accutase (BioLegend). Single primary sphere cells were subjected to sphere culture for secondary sphere formation in a same way as primary sphere formation assays. The criteria for secondary sphere counting are the same as those for primary sphere counting. When secondary sphere formation involves siRNA knockdown, primary sphere cells were cultured in the sphere culture medium containing siRNA-lipofectamine complexes for continuous gene knockdown.

## Fluorescence-activated cell sorting (FACS) analysis

Cells were stained with the following antibodies from Biolegend (San Diego, CA): allophycocyanin (APC)-conjugated anti-CD44 (RRID:AB_312963), phycoerythrin (PE)-conjugated anti-CD24 (RRID:AB_314854) and APC-conjugated anti-EpCAM (RRID:AB_756081). FACS analysis of stained cells was performed using a FACSAria II cell sorter (Becton Dickinson, Franklin Lakes, NJ) as previously described (*Lo et al., 2016*).

## Immunofluorescent assay

Immunofluorescent staining analysis of E-cadherin and vimentin was performed as previously described (*Lo et al., 2017*). Antibodies used in IF assays include mouse monoclonal anti-human E-cadherin (RRID:AB_1955688) and anti-human vimentin (RRID:AB_2537032) Abs.

## *In vivo* tumorigenicity assay

The control MDA-MB-468 and its derived stable cell lines ($1 \times 10^6$ or $5 \times 10^6$ cells in 50% Matrigel) were injected into the fourth mammary fat pads of 6 weeks old female athymic nude mice (n = 6 for each cell line). Athymic nude mice (NU/NU mice, Stock No: 002019) were obtained from the Jackson

Laboratory (Bar Harbor, ME). The length and width of tumors were measured every four days with a caliper to calculate tumor volume using the formula: V = 1/2 (Length $\times$ Width$^2$) (*Jensen et al., 2008*). At the endpoint, xenograft tumors were isolated and processed for IHC analysis. Xenograft tumor experiments were performed according to the animal protocol approved by the Institutional Animal Care and Use Committee (IACUC) of the University of Maryland School of Medicine, which is in accordance with the guidelines established by the USPHS.

### In vivo metastasis assay

For *in vivo* metastasis analysis of TNBC cells, we performed tail vein injection of $1 \times 10^6$ MDA-MB-468 cells expressing either scramble or LIPG shRNA into athymic nude mice. Four weeks after injection, experimental mice were euthanized and dissected for lung isolation. Isolated lung were fixed and embedded in paraffin for making tissue sections. Paraffin-embedded lung tissue sections were stained with hematoxylin and eosin (HE staining) for histological analysis of metastatic tumor foci. IHC analysis of vimentin protein expression in lung tumor foci was also performed using the anti-vimentin antibody (Proteintech, Rosemont, IL) to confirm metastasis of MDA-MB-468. The numbers of lung tumor foci were counted according to HE and IHC staining of lung tissue samples. *In vivo* metastasis experiments were performed according to the animal protocol approved by the Institutional Animal Care and Use Committee (IACUC) of the University of Maryland School of Medicine, which is in accordance with the guidelines established by the USPHS.

### Proteomic profiling analysis

Parental control and LIPG-overexpressing MCF7 cells were lysed with 4% sodium deoxycholate. Lysates were washed, reduced, alkylated and trypsinolyzed in filter as described by Wisniewski et al. (*Wiśniewski et al., 2009*) and Erde et al. (*Erde et al., 2014*). Peptides were separated on a nano-ACQUITY UPLC analytical column (BEH130 C18, 1.7 μm, 75 μm x 200 mm, Waters) over a 180 min linear acetonitrile gradient (3–43%) with 0.1% formic acid on a Waters nano-ACQUITY UPLC system, and analyzed on a coupled Waters Synapt G2S HDMS mass spectrometric system. Spectra were acquired using an ion mobility linked parallel mass spectrometry (UDMSe) and analyzed as described by *Distler et al. (2014)*.

Peaks were resolved using Apex3D and Peptide3D algorithms (*Geromanos et al., 2009*). Tandem mass spectra were searched against a UniProt human reference proteome and its corresponding decoy sequences using an ion accounting algorithm (*Li et al., 2009*). Resulting hits were validated at a maximum false discovery rate of 0.04. Abundance ratios between the LIPG overexpressing cell line and the control cell line were measured by comparing the MS1 peak volumes of peptide ions at the low collision energy cycle, whose identities were confirmed by MS2 sequencing at the elevated collision energy cycle as described above. Label-free quantifications were performed using an aligned AMRT (Accurate Mass and Retention Time) cluster quantification algorithm developed by *Qi et al. (2012)*.

### Statistical analysis

Data are presented as mean ± S.D. Statistical analysis of general experimental datasets was performed by Student's t test. Statistical analysis of tumor growth curves was performed by two-way ANOVA. F-test was used to analyze variances between two gene expression datasets from the Oncomine database. Correlation between mRNA levels of two genes was analyzed by Pearson correlation test. The p values of <0.05 were considered significant. Data were analyzed using the GraphPad Prism software (version 6.0; GraphPad Software, Inc, La Jolla, CA) (RRID:SCR_002798).

## Acknowledgements

We thank Yan-Jen Lo for technical assistance with *in silico* data analysis, Justine E Yu and Benjamin Wolfson for assistance with manuscript editing. This work was supported, in whole or in part, by the NIH CA157779A1 and CA163820A1 to Q Zhou, and the University of Maryland Baltimore, School of Pharmacy Mass Spectrometry Center (SOP1841-IQB2014) to W Huang and MA Kane.

## Additional information

### Funding

| Funder | Grant reference number | Author |
|---|---|---|
| University of Maryland, Baltimore County | SOP1841-IQB2014 | Weiliang Huang<br>Maureen A Kane |
| National Cancer Institute | CA157779A1 | Qun Zhou |
| National Cancer Institute | CA163820A1 | Qun Zhou |

The funders had no role in study design, data collection and interpretation, or the decision to submit the work for publication.

### Author contributions

Pang-Kuo Lo, Conceptualization, Data curation, Formal analysis, Validation, Investigation, Visualization, Methodology, Writing—original draft, Writing—review and editing; Yuan Yao, Data curation, Formal analysis, Methodology, Project administration; Ji Shin Lee, Resources, Visualization, Methodology, Project administration, Writing—review and editing; Yongshu Zhang, Data curation, Methodology; Weiliang Huang, Data curation, Funding acquisition, Methodology, Writing—review and editing; Maureen A Kane, Resources, Data curation, Funding acquisition, Methodology, Writing—review and editing; Qun Zhou, Conceptualization, Resources, Supervision, Funding acquisition, Visualization, Project administration, Writing—review and editing

### Author ORCIDs

Pang-Kuo Lo (iD) http://orcid.org/0000-0001-7202-3162
Qun Zhou (iD) http://orcid.org/0000-0003-1745-0369

### Ethics

Human subjects: Breast cancer tissue samples from breast cancer patients were provided by the Chonnam National University Hwasun Hospital National Biobank of Korea, a member of the National Biobank of Republic of Korea, which is supported by the Ministry of Health, Welfare and Family Affairs. All tissue samples were obtained with informed consent from patients under protocols approved by the institutional review board of the Chonnam National University Hwasun Hospital. The use of human tissue specimens in this study has been approved by the institutional review board of the Chonnam National University Hwasun Hospital (Reference number: CNUHH-2016-153). The institutional approval is not required for publication of data from these human specimens due to institutional policies of the Chonnam National University Hwasun Hospital.
Animal experimentation: This animal study was performed in strict accordance with the recommendations in the Guide for the Care and Use of Laboratory Animals of the National Institutes of Health. All of the animals were handled according to approved institutional animal care and use committee (IACUC) protocols (#0116028) of the University of Maryland School of Medicine.

### Decision letter and Author response

Decision letter https://doi.org/10.7554/eLife.31334.042
Author response https://doi.org/10.7554/eLife.31334.043

## Additional files

### Supplementary files

• Transparent reporting form
DOI: https://doi.org/10.7554/eLife.31334.040

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
