## [Decision Letter]

Thank you for submitting your article "The novel DTX3L-LIPG-ISG15 signaling axis promotes the development of basal-like triple-negative breast cancer" for consideration by *eLife*. Your article has been favorably evaluated by Charles Sawyers (Senior Editor) and three reviewers, one of whom, Ralph DeBerardinis (Reviewer #1), is a member of our Board of Reviewing Editors.

The reviewers have discussed the reviews with one another and the Reviewing Editor has drafted this decision to help you prepare a revised submission.

Summary:

These authors studied the role of lipoprotein lipases in breast cancer and identify endothelial lipase (LIPG) as being over-expressed in basal-like triple negative breast cancers and cell lines. Many of the aggressive features of these cells, including proliferation, propensity to form spheroids, migration/invasion, expression of genes related to the epithelial-mesenchymal transition (EMT) and tumor growth in mice, require LIPG expression. The authors use proteomics to demonstrate that many of the proteins induced though LIPG over-expression are interferon-stimulating genes. One of these, ISG15, is then shown to execute many of LIPG's effects on the basal-like phenotype. The authors also show that LIPG levels are regulated by DTX3L via effects on protein stability. Over-expressing DTX3L is sufficient to drive an EMT-like phenotype and promote tumor growth *in vivo*, but these effects are reversed by silencing LIPG or ISG15. The authors conclude that the DTX3L-LIPG-ISG15 signaling axis confers key aspects of the basal-like TNBC phenotype and promotes tumor growth.

Essential revisions:

1) The authors should do more to strengthen the mechanistic link between LIPG and its downstream targets, particularly ISG15. This is important because the rationale for studying lipase function in TNBC in the first place was somewhat non-intuitive. Although this would not fully explain the mechanistic basis of LIPG's role in the reported pathway, one basic and essential question is whether the catalytic activity of LIPG (i.e. its metabolic function) is relevant for the regulation of ISG15. To answer this question, the authors could overexpress an allele of LIPG with a point mutation disrupting its lipase activity in MCF7 cells, then examine whether this mutant confers the same changes in gene expression, migration, etc. observed with wild-type LIPG.

2) Figure 1 needs to include a larger panel of cell lines to better validate the conclusions about the importance of LIPG in the basal subset. For example, the panel currently only includes two cell lines with the luminal phenotype, and this is inadequate to draw any conclusions.

3) The role of the DTX3L-LIPG-ISG15 signaling pathway in stem cell and EMT phenotypes should be supported by additional evidence. For stem cell properties, the authors should assess the effect of silencing multiple components of the pathway on secondary sphere formation, which should more robustly test stem-like properties. EMT should be assessed through a more thorough evaluation of morphological changes and immunostaining/immunofluorescence of classical EMT markers like vimentin and E-cadherin.

4) As metastasis is an essential feature of TNBC aggressiveness, it would strengthen the paper considerably if LIPG silencing reduced MDA-MB-468 metastasis *in vivo*.

---

## [Author Response]

Essential revisions:1) The authors should do more to strengthen the mechanistic link between LIPG and its downstream targets, particularly ISG15. This is important because the rationale for studying lipase function in TNBC in the first place was somewhat non-intuitive. Although this would not fully explain the mechanistic basis of LIPG's role in the reported pathway, one basic and essential question is whether the catalytic activity of LIPG (i.e. its metabolic function) is relevant for the regulation of ISG15. To answer this question, the authors could overexpress an allele of LIPG with a point mutation disrupting its lipase activity in MCF7 cells, then examine whether this mutant confers the same changes in gene expression, migration, etc. observed with wild-type LIPG.

We appreciate this very constructive comment. As suggested, we created a MCF7 line expressing a catalytically inactive LIPG mutant protein (S149A) that has been shown to lose the lipase function (JBC, 2003, 278: 40688–40693). We conducted a series of studies of this mutant line in comparison to parental and wild-type LIPG-overexpressing MCF7 lines to examine the role of LIPG catalytic lipase activity in the basal/EMT features and cancer stem cell phenotypes. As shown in Figure 3, although LIPG^S149A^-expressing MCF7 cells had a lower proliferation rate and formed smaller cancer stem cell spheres compared to parental and wild-type LIPG-expressing MCF7 cells, they showed increased sphere formation compared to parental MCF7 cells. Moreover, similar to wild-type LIPG, LIPG^S149A^ was also able to promote the expression of stem cell genes and increase the CD24-CD44+ CSC population. With regard to the basal/EMT features, LIPG^S149A^ was potent to induce expression of cytokeratin 14 (a basal/myoepithelial protein) and ZEB1 (an EMT gene), downregulate expression of E-cadherin and EpCAM (a luminal marker protein), and promote cell migration to an equivalent or greater extent when compared to wild-type LIPG. These results suggest that the lipase function of LIPG plays a significant role in promoting the cancer cell proliferation rate, but may not be essential for stemness and EMT phenotypes of cancer cells. Consistent with these findings, LIPG^S149A^ was competent to induce the increased expression of ISG15 in MCF7 cells to a similar level as its wild-type counterpart (see Figure 6). These findings together suggest that LIPG possesses both enzymatic and non-enzymatic functions that regulate the different aspects of cancer malignancies. These results are stated in the "Results" section (see the subsection “LIPG-mediated enhancements of basal/EMT, migratory and cancer stem cell features are largely independent of its lipase catalytic activity”) and discussed in the "Discussion" section (see the second paragraph).

2) Figure 1 needs to include a larger panel of cell lines to better validate the conclusions about the importance of LIPG in the basal subset. For example, the panel currently only includes two cell lines with the luminal phenotype, and this is inadequate to draw any conclusions.

As suggested, we have included the LIPG expression data of 48 breast cancer cell lines in Figure 1 of the revised manuscript. This BC line cohort includes basal-like (n=20), Her2-amplified (n=15) and luminal (n=13) breast cancer cell lines. Consistent with the LIPG expression data of breast cancer tissue specimens (Figure 1), LIPG was expressed at a higher level in TNBC/basal-like cell lines compared to HER2-amplified and luminal breast cancer cell lines.

3) The role of the DTX3L-LIPG-ISG15 signaling pathway in stem cell and EMT phenotypes should be supported by additional evidence. For stem cell properties, the authors should assess the effect of silencing multiple components of the pathway on secondary sphere formation, which should more robustly test stem-like properties. EMT should be assessed through a more thorough evaluation of morphological changes and immunostaining/immunofluorescence of classical EMT markers like vimentin and E-cadherin.

As suggested, in addition to primary sphere formation data, we have included the secondary sphere formation data from multiple experiments to reveal the role of the DTX3L-LIPG-ISG15 signaling pathway in cancer stem cells (please see Figure 2, Figure 3, Figure 4, Figure 5, Figure 6, and 8H). Consistent with the primary sphere data, knockdown of either DTX3L or LIPG had a significant inhibitory effect on the secondary sphere formation of TNBC cells, whereas LIPG overexpression promoted the secondary sphere formation of MCF7 cells. Moreover, ISG15 knockdown moderately suppressed the secondary formation of TNBC cells. This result is in line with our previous conclusion according to the primary sphere data; ISG15 plays a more prominent role in the EMT than in cancer stem cells.

To strengthen the role of the DTX3L-LIPG-ISG15 signaling axis in modulating EMT phenotypes, we performed immunostaining of E-cadherin and vimentin in multiple experiments. We have included these results in Figure 2, Figure 3, Figure 4, Figure 6, Figure 8, Figure 5—figure supplement 2, and Figure 6—figure supplement 3. Our immunostaining data have shown that the DTX3L-LIPG-ISG15 signaling axis is critical for the EMT phenotypes of TNBC cells as this pathway promotes vimentin protein expression and inhibits E-cadherin protein expression. These results are consistent with the data of EMT gene expression, flow cytometry analysis of EpCAM, migration and invasion experiments. Moreover, we observed that the DTX3L-LIPG-ISG15 signaling pathway is required for the mensenchymal morphology of MDA-MB-468 as its inactivation led to a morphological shift to an epithelial-like phenotype (please see Figure 5—figure supplement 2 and Figure 5—figure supplement 3, Figure 6—figure supplement 3 and Figure 8).

4) As metastasis is an essential feature of TNBC aggressiveness, it would strengthen the paper considerably if LIPG silencing reduced MDA-MB-468 metastasis *in vivo*.

We performed a tail vein injection experiment to examine the impact of LIPG knockdown on *in vivo* lung metastasis of MDA-MB-468 in nude mice. As shown in Figure 5, LIPG knockdown completely inhibited MDA-MB-468 metastasis to the lung. This result suggests that LIPG is functionally required for TNBC metastasis *in vivo*. The result has been included in the revised manuscript (see the subsection “LIPG is required for the tumorigenicity and metastasis of basal-like TNBC”, the last paragraph).